# Neural Event-Triggered Control with Optimal Scheduling

**Luan Yang** [* 1]  **Jingdong Zhang** [* 1 2]  **Qunxi Zhu** [1 3 4]  **Wei Lin** [1 2 3 4]

## Abstract

Learning-enabled controllers with stability certificate functions have demonstrated impressive empirical performance in addressing control problems in recent years. Nevertheless, directly deploying the neural controllers onto actual digital platforms requires impractically excessive communication resources due to a continuously updating demand from the closed-loop feedback controller. We introduce a framework aimed at learning the event-triggered controller (ETC) with optimal scheduling, i.e., minimal triggering times, to address this challenge in resource-constrained scenarios. Our proposed framework, denoted by Neural ETC, includes two practical algorithms: the path integral algorithm based on directly simulating the event-triggered dynamics, and the Monte Carlo algorithm derived from new theoretical results regarding lower bound of inter-event time. Furthermore, we propose a projection operation with an analytical expression that ensures theoretical stability and schedule optimality for Neural ETC. Compared to the conventional neural controllers, our empirical results show that the Neural ETC significantly reduces the required communication resources while enhancing the control performance in constrained communication resources scenarios.

## 1. Introduction

Stabilizing the complex nonlinear systems represents a formidable focal task within the realms of mathematics and

---
[*]Equal contribution [1]Research Institute of Intelligent Complex Systems, Fudan University, China. [2]School of Mathematical Sciences, LMNS, and SCMS, Fudan University, China. [3]State Key Laboratory of Medical Neurobiology and MOE Frontiers Center for Brain Science, Institutes of Brain Science, Fudan University, China. [4]Shanghai Artificial Intelligence Laboratory, China. Correspondence to: Qunxi Zhu <qxzhu16@fudan.edu.cn>, Wei Lin <wlin@fudan.edu.cn>.

*Proceedings of the 42$^{nd}$ International Conference on Machine Learning*, Vancouver, Canada. PMLR 267, 2025. Copyright 2025 by the author(s).

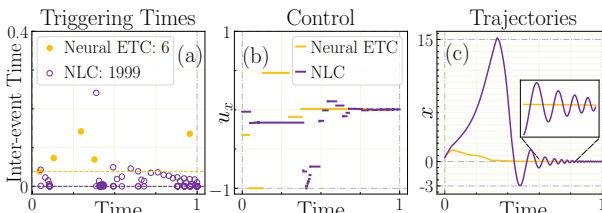

*Figure 1.* Comparison of Neural ETC (yellow) and neural Lyapunov control (NLC, purple) in stabilizing the Lorenz system under the event-triggered control setting. (a): The inter-event time of consecutive triggering events. The dashed lines represent the minimal inter-event time of each control. (b): The control values acting on variable $x$ in the control process. (c): The controlled trajectories of the variable $x$.

engineering. Previous research in the field of cybernetics has applied the Lyapunov stability theory to formulate stabilizing policies for linear or polynomial dynamical systems, including the linear quadratic regulator (LQR) (Khalil, 2002) and the sum-of-squares (SOS) polynomials, using the semi-definite planning (SDP) (Parrilo, 2000). Stabilizing more intricate dynamical systems with high dimension and nonlinearity, as encountered in real applications, has prompted the integration of machine learning techniques into the cybernetics community(Tsukamoto et al., 2021). Recent advancements in learning neural networks based controllers with certificate functions, such as Lyapunov function (Chang et al., 2019; Zhang et al., 2022a), LaSalle function (Zhang et al., 2022b), barrier functions (Qin et al., 2020) and contraction metrics (Sun et al., 2021), have demonstrated outstanding performance in controlling diverse dynamics (Dawson et al., 2022). Nevertheless, it is noteworthy that all these controllers require updating the control signal continuously over time, leading to a considerable communication cost between controller and platform.

The periodic control is mostly advocated for implementing feedback control laws on digital platforms (Franklin et al., 2002). However, such implementations often incur significant over-provisioning of the communication network, especially in the recently developed large-scale resource-constrained wireless embedded control systems (Lemmon, 2010). To mitigate this issue, event-triggering mechanism is introduced to generate sporadic transmissions across the feedback channels of the system. Compared to periodic

control which updates the control signal at a series of pre-defined explicit times, event-triggered control updates the control signal at the instants when the current measurement violates a predefined triggering condition, thereby triggering a state-dependent event (Heemels et al., 2012). Given that these instants are implicitly determined by the state trajectories, the scheduling of computation and communication resources for event-triggered control becomes a very challenging problem, involving the minimization of the number of events and the increase of inter-event time. While significant strides have been made in stabilizing specific dynamics with event-triggered control in recent years, the task of designing event-triggered control for general nonlinear and large-scale dynamics with optimal scheduling remains an open problem (Åarzén, 1999; Tabuada, 2007; Heemels et al., 2008; Henningsson et al., 2008).

Our goal is to design event-triggered control for general complex dynamics, ensuring both stability guarantee and optimal scheduling, i.e., to implement event-triggered control with the minimal triggering times and the maximal inter-event time. Fig. 1 depicts the comparison of control performance of the Neural ETC and the NLC in the event-triggered realization to stabilize a Lorenz dynamic. In Fig. 1(a)-1(b), it is evident that the triggering times of Neural ETC are significantly fewer than those of NLC, and the minimal inter-event time of consecutive triggering times of Neural ETC considerably exceeds that of NLC. These disparities lead to the different behaviors of the controlled trajectories, as depicted in Fig. 1(c).Under Neural ETC, the trajectory rapidly converges to the target state, while the NLC exhibits violent oscillation around the target.

**Contribution.** The principal contributions of this paper can be summarized as follows:

- We propose Neural ETC, a framework for learning event-triggered controllers ensuring both stability guarantee and optimal scheduling, where the exponential stability comes from the devised event function.

- Specifically, we firstly propose a path integral approach to realize the implementation of the machine learning framework based on the root solver and neural event ODE solver that calculate the trainable event triggering times. Secondly, we theoretically address the estimation of the minimal inter-event time of the event triggered controlled system, which leading to the Monte Carlo approach of our framework that circumvents the expensive computation cost of back-propagation through ODE solvers. The two approaches trade off in terms of stabilization performance and training efficiency, which is convenient for users to flexibly choose the specific approach according to the task in hand.

- To theoretically guarantee the stability of the controlled

system under our Neural ETC, we propose the projection operation that rigorously endows our Neural ETC with stability and schedule optimality. Instead of solving an optimization problem to obtain the projection, we provide analytical expression for our projection operation, leading to a fast implementation for our framework.

- Finally, we evaluate Neural ETCs on a variety of representative physical and engineering systems. Compared to existing stabilizing controllers, we find that Neural ETCs exhibit significant superiority in decreasing the triggering times and maximizing the minimal inter-event time. The code for reproducing all the numerical experiments is released at https://github.com/jingddong-zhang/Neural-Event-triggered-Control (hyperlink of Neural ETC).

## 2. Background

**Notations.** Denote by $\|\cdot\|$ the $L^2$-norm for any given vector in $\mathbb{R}^d$. Denote by $\|\cdot\|_{C(\mathcal{D})}$ the maximum norm on continuous function space $C(\mathcal{D})$. For $A = (a_{ij})$, a matrix of dimension $d \times r$, denote by $\|A\|_{\mathrm{F}}^2 = \sum_{i=1}^{d} \sum_{j=1}^{r} a_{ij}^2$ the Frobenius norm. Denote $\max(a, 0)$ by $(a)^+$. Denote $\boldsymbol{x} \cdot \boldsymbol{y}$ as the inner product of two vectors.

### 2.1. Neural Lyapunov Control

To begin with, we consider the feedback controlled dynamical system of the following general form:

$$\dot{\boldsymbol{x}} = \boldsymbol{f}(\boldsymbol{x}, \boldsymbol{u}(\boldsymbol{x})) \triangleq \boldsymbol{f}_{\boldsymbol{u}}(\boldsymbol{x}), \ \boldsymbol{x} \in \mathbb{R}^d, \ \boldsymbol{u} \in \mathbb{R}^m, \quad (1)$$

where $\boldsymbol{f}_{\boldsymbol{u}}(\boldsymbol{x}) : \mathcal{D} \to \mathbb{R}^d$ is the Lipschitz-continuous vector field acting on some prescribed open set $\mathcal{D} \subset \mathbb{R}^d$. The solution initiated at time $t_0$ from $\boldsymbol{x}_0$ under controller $\boldsymbol{u}$ is denoted by $\boldsymbol{x}_{\boldsymbol{u}}(t; t_0, \boldsymbol{x}_0)$. For brevity, we let the unstable equilibrium $\boldsymbol{x}^* \in \mathcal{D}$ be origin, i.e., $\boldsymbol{f}(\boldsymbol{0}, \boldsymbol{0}) = \boldsymbol{0}$. One major problem in cybernetics field is to design stabilizing controller $\boldsymbol{u}(\boldsymbol{x})$ (Wiener, 2019) such that $\lim_{t \to \infty} \boldsymbol{x}_{\boldsymbol{u}}(t; t_0, \boldsymbol{x}_0) = \boldsymbol{0}$, for any initial value $\boldsymbol{x}_0 \in \mathcal{D}$.

**Theorem 2.1.** *(Mao, 2007) Suppose there exists a continuously differentiable function $V : \mathcal{D} \to R$ that satisfies the following conditions:* (i) $V(0) = 0$, (ii) $V(\boldsymbol{x}) \geq c\|\boldsymbol{x}\|^p$ *for some constants $c, p > 0$,* (iii) *and $\mathcal{L}_{\boldsymbol{f}_{\boldsymbol{u}}} V < -\delta V$, for some $\delta > 0$.* [1] *Then, the system is exponentially stable at the origin, that is,* $\limsup_{t \to \infty} \frac{1}{t} \log \|\boldsymbol{x}_{\boldsymbol{u}}(t; t_0, \boldsymbol{x}_0)\| \leq -\frac{\delta}{p}$. *Here $V$ is called a Lyapunov function.*

Previous works parameterize the controller and the Lyapunov function as $\boldsymbol{u}_{\boldsymbol{\phi}}, V_{\boldsymbol{\theta}}$, and integrate the sufficient con-

---

[1] $\mathcal{L}_{\boldsymbol{f}_{\boldsymbol{u}}} V$ represent the Lie derivative of $V$ along the direction $\boldsymbol{f}_{\boldsymbol{u}}$, i.e., $\mathcal{L}_{\boldsymbol{f}_{\boldsymbol{u}}} V = \nabla V \cdot \boldsymbol{f}_{\boldsymbol{u}}$.

ditions (i)-(iii) in Theorem 2.1 for Lyapunov stability into the loss function (Chang et al., 2019; Zhang et al., 2022a; Dawson et al., 2023). The learned Lyapunov $V_{\theta}$ plays a role of stability certificate function.

*Remark* 2.2. Unlike model-free reinforcement learning (RL) approaches that search for an online control policy guided by a reward function along the trajectories of the dynamical systems. (Kaelbling et al., 1996), the neural Lyapunov control searches for an offline policy and a certificate function $V$ that proves the soundness of the learned policy (Dawson et al., 2022). Nevertheless, updating the feedback policy continuously in the implementation process incurs prohibitive high communication cost.

## 2.2. Event-triggered Control

Although the feedback controller works well in numerical simulations, updating and implementing the controller continuously is impractical in most real-world digital platforms under communication constraints (Åström & Bernhardsson, 1999). To conquer this weakness, event-triggered stabilizing control is introduced as follows (Heemels et al., 2012),

**Definition 2.3.** (Event-triggered Control) Consider the controlled system (1), the event-triggered controller is defined as $\boldsymbol{u}(t) = \boldsymbol{u}(\boldsymbol{x}(t_k))$, $t_k \leq t < t_{k+1}$, where the triggering time is decided by $t_{k+1} = \inf\{t > t_k : h(\boldsymbol{x}(t)) = 0\}$ for some predefined event function $h$. The largest lower bound $\tau^*$ of $\{t_{k+1} - t_k\}$ is called as minimal inter-event time. For example, if there exists a Lyapunov function $V$ for the feedback controlled system (1), then the event function is set to guarantee the Lyapunov condition on each event triggering time interval, i.e., $\nabla V \cdot \boldsymbol{f}(\boldsymbol{x}(t), \boldsymbol{u}(\boldsymbol{x}_{t_k})) < 0$, $t \in [t_k, t_{k+1})$.

**Problem Statement.** We assume that the zero solution of the uncontrolled system in Eq. (1) is unstable, i.e. $\lim_{t \to \infty} \boldsymbol{x}_{\boldsymbol{u}=\boldsymbol{0}}(t; t_0, \boldsymbol{x}_0) \neq \boldsymbol{0}$. We aim at stabilizing the zero solution using event-triggered control based on neural networks (NNs) with optimal scheduling, i.e., the least triggering times, which is urgently required by the digital platforms wherein the communication resources of updating the control value are limited. Notice that in an average sense, the triggering times are inversely proportional to the inter-event time, our goal is equivalently to leverage the NNs to design an appropriate controller $\boldsymbol{u}$ with $\boldsymbol{u}(\boldsymbol{0}) = \boldsymbol{0}$ such that the controlled system under event-triggered implementation is steered to the zero solution with the maximal inter-event time. We summarize the problem formulation as the following optimization problem, where the triggering time $\{t_k : t_k \leq T\}$ depends on the controller $\boldsymbol{u}$ and the triggering mechanism, and $T \leq \infty$ is the prefixed time limit according to the specific tasks. We aim at devising controller $\boldsymbol{u}$ and triggering mechanism to solve this problem

based on the known model $\boldsymbol{f}$ and time limit $T$.

$$\min_{\boldsymbol{u}} \left( \frac{1}{\min_{\{t_k \leq T\}}(t_{k+1} - t_k)} \right) + \lambda_1 \|\boldsymbol{u}(\boldsymbol{x})\|_{C(\mathcal{D})}$$

$$\text{s.t.} \quad \dot{\boldsymbol{x}}(t) = \boldsymbol{f}(\boldsymbol{x}(t), \boldsymbol{u}(\boldsymbol{x}(t_k))), \ t \in [t_k, t_{k+1}),$$

$$\boldsymbol{x}(0) = \boldsymbol{x}_0 \in \mathcal{D}, \lim_{t \to T} \boldsymbol{x}(t) = \boldsymbol{0}, \tag{2}$$

The major difficulty of this problem comes from that the implicitly defined triggering times are not equidistant, and are only known when the events are triggered (Miskowicz, 2018). The majority of existing works focus on the stabilization performance of event-triggered control and often omit the communication cost of updating the control value at triggering moments. In what follows, we propose neural event-triggered control (Neural ETC) framework to address both the stabilization and the communication cost issues of event-triggered control.

## 3. Method

**Closed-loop controlled dynamics.** The dynamics under event-triggered control is generally an open-loop system with controller varying from different triggering time intervals. In order to simplify the theoretical analysis and to utilize the existing numerical tools for ODE solvers, we transform the event-triggered controlled system to the closed-loop version via augmenting the dynamics with an error state $\boldsymbol{e}(t) = \boldsymbol{x}(t_k) - \boldsymbol{x}(t)$, $t \in [t_k, t_{k+1})$ and an update operation $\boldsymbol{e}(t_{k+1}) = \boldsymbol{0}$. Then we obtain the closed-loop controlled dynamics as

$$\dot{\boldsymbol{x}} = \boldsymbol{f}(\boldsymbol{x}, \boldsymbol{u}(\boldsymbol{x} + \boldsymbol{e})), \dot{\boldsymbol{e}} = -\boldsymbol{f}(\boldsymbol{x}, \boldsymbol{u}(\boldsymbol{x} + \boldsymbol{e})), \ t \in [t_k, t_{k+1}).$$

In the next sections, we construct the event function with exponential stability guarantee and deduce the theoretical estimation of minimal inter-event time based on the augmented dynamics of $(\boldsymbol{x}, \boldsymbol{e})$.

**Event function for exponential stability.** We consider the exponential Lyapunov stability for controlled system (1) such that the corresponding Lyapunov function defined in Theorem 2.1 satisfies the stability condition $\mathcal{L}_{\boldsymbol{f}_u} V \leq -\delta V$ and $V(\boldsymbol{x}) \geq \alpha(\|\boldsymbol{x}\|)$, where $\alpha$ is a class-$K$ function[2]. For brevity, we fix $\delta = 1$ in this paper such that the decay exponent of the Lyapunov function is $1$. Since the event-triggered controller is a discrete time realization of the original feedback controller $\boldsymbol{u}$, the corresponding exponential decay rate of the Lyapunov function is less than $1$. Therefore, we design the event function $h = h(\boldsymbol{x}, \boldsymbol{e})$ as

$$h = \nabla V(\boldsymbol{x}) \cdot (\boldsymbol{f}(\boldsymbol{x}, \boldsymbol{u}(\boldsymbol{x} + \boldsymbol{e})) - \boldsymbol{f}(\boldsymbol{x}, \boldsymbol{u}(\boldsymbol{x}))) - \sigma V(\boldsymbol{x}) \tag{3}$$

---

[2]A continuous function $\alpha : (0, \infty) \to (0, \infty)$ is said to belong to class-$K$ if it is strictly increasing and $\alpha(0) = 0$.

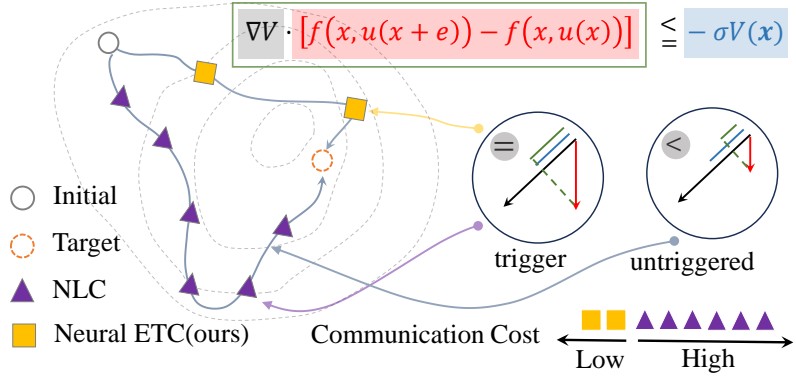

*Figure 2.* Illustration of the Neural ETC with optimal scheduling.

with $0 < \sigma < 1$, such that the event-triggered controlled system satisfies

$$\nabla V(\boldsymbol{x}) \cdot \boldsymbol{f}(\boldsymbol{x}, \boldsymbol{u}(\boldsymbol{x} + \boldsymbol{e})) \leq \nabla V(\boldsymbol{x}) \cdot \boldsymbol{f}(\boldsymbol{x}, \boldsymbol{u}(\boldsymbol{x})) + \sigma V(\boldsymbol{x})$$
$$\leq -(1 - \sigma)V(\boldsymbol{x}). \tag{4}$$

Hence, the exponential stability of the event-triggered controlled system is assured with exponential decay rate $1 - \sigma$. As illustrated in Fig. 2, the NLC is used as an example method to be compared with the Neural ETC. The control value is updated when an event is triggered, i.e., the event function $h$ equals to zero. Our method achieves the exponential stability and has the least triggered events, leading to the lowest communication cost of updating the control value.

### 3.1. Path Integral Approach

**Parameterization.** In order to design the feedback controller such that its event-triggered implementation stabilize the unstable equilibrium efficiently and has the largest minimal inter-event time, we consider the following parameterized optimization problem.

$$\min_{\boldsymbol{\theta}, \boldsymbol{\phi}} \left( \min_{\{t_k \leq T\}} \frac{1}{t_{k+1} - t_k} \right) + \lambda_1 \|\boldsymbol{u}_{\boldsymbol{\phi}}(\boldsymbol{x})\|_{C(\mathcal{D})}$$

$$\text{s.t.} \quad \dot{\boldsymbol{x}} = \boldsymbol{f}(\boldsymbol{x}, \boldsymbol{u}_{\boldsymbol{\phi}}(\boldsymbol{x} + \boldsymbol{e})), \ t \in [t_k, t_{k+1}),$$
$$\dot{\boldsymbol{e}} = -\boldsymbol{f}(\boldsymbol{x}, \boldsymbol{u}_{\boldsymbol{\phi}}(\boldsymbol{x} + \boldsymbol{e})), \ t \in [t_k, t_{k+1}),$$
$$\boldsymbol{x}(0) = \boldsymbol{x}_0, \ \boldsymbol{e}(t_k) = \boldsymbol{0}, \ V_{\boldsymbol{\theta}}(\boldsymbol{0}) = 0, \ \boldsymbol{u}_{\boldsymbol{\phi}}(\boldsymbol{0}) = \boldsymbol{0},$$
$$t_{k+1} = \inf_{t > t_k} \{t : h_{\boldsymbol{\theta}, \boldsymbol{\phi}}(\boldsymbol{x}(t), \boldsymbol{e}(t)) = 0\}$$
$$\alpha(\|\boldsymbol{x}\|) - V_{\boldsymbol{\theta}}(\boldsymbol{x}) \leq 0, \ \mathcal{L}_{\boldsymbol{f}_{\boldsymbol{u}_{\boldsymbol{\phi}}}} V_{\boldsymbol{\theta}}(\boldsymbol{x}) + V_{\boldsymbol{\theta}}(\boldsymbol{x}) \leq 0.$$

Here, $\lambda_1$ is a predefined weight factor, $T$ is the temporal length of the controlled trajectory, $\alpha$ is a class-$K$ function, and $h_{\boldsymbol{\theta}, \boldsymbol{\phi}}(\boldsymbol{e}, \boldsymbol{x}) = \mathcal{L}_{\boldsymbol{f}_{\boldsymbol{u}_{\boldsymbol{\phi}}}} V_{\boldsymbol{\theta}} \cdot (\boldsymbol{f}(\boldsymbol{x}, \boldsymbol{u}_{\boldsymbol{\phi}}(\boldsymbol{x} + \boldsymbol{e})) - \boldsymbol{f}(\boldsymbol{x}, \boldsymbol{u}_{\boldsymbol{\phi}}(\boldsymbol{x}))) - \sigma V_{\boldsymbol{\theta}}(\boldsymbol{x}(t))$ is the parameterized event function. To ensure the neural functions $V_{\boldsymbol{\theta}}, \boldsymbol{u}_{\boldsymbol{\phi}}$ satisfy some constraints naturally, we adopt the

parametrization in (Zhang et al., 2022a) as follows,

$$V_{\boldsymbol{\theta}} = \text{ICNN}_{\boldsymbol{\theta}}(\boldsymbol{x}) - \text{ICNN}_{\boldsymbol{\theta}}(\boldsymbol{0}) + \varepsilon \|\boldsymbol{x}\|^2,$$
$$\boldsymbol{u}_{\boldsymbol{\phi}} = \text{diag}(\boldsymbol{x}) \text{NN}_{\boldsymbol{\phi}}(\boldsymbol{x}) \text{ or } \text{NN}_{\boldsymbol{\phi}}(\boldsymbol{x}) - \text{NN}_{\boldsymbol{\phi}}(\boldsymbol{0}), \tag{5}$$

where $\text{diag}(\boldsymbol{x})$ transforms a vector to a diagonal matrix with $(\text{diag}(\boldsymbol{x}))_{ij} = \delta_{ij}x_i$, $\text{ICNN}_{\boldsymbol{\theta}}$ and $\text{NN}_{\boldsymbol{\phi}}$ represent the input convex neural network and the feedforward neural networks, respectively, the detailed formulation is provided in Appendix A.3.1. We minimize the continuous function norm $\|\boldsymbol{u}_{\boldsymbol{\phi}}\|_{C(\mathcal{D})}$ by regularizing the Lipschitz constant of the neural network, we apply the spectral norm regularization method in (Yoshida & Miyato, 2017) to minimize the spectral norm of the weight matrices $\{\boldsymbol{W}_{\boldsymbol{\phi}, i}\}_{i=1}^l$ in $\boldsymbol{u}_{\boldsymbol{\phi}}$ with the regularization term $L_{\text{lip}} = \sum_{i=1}^l \sigma(\boldsymbol{W}_{\boldsymbol{\phi}, i})^2$. To solve the substantially non-convex optimization problem, we relax the original hard constraint $\mathcal{L}_{\boldsymbol{f}_{\boldsymbol{u}_{\boldsymbol{\phi}}}} V_{\boldsymbol{\theta}}(\boldsymbol{x}) + V_{\boldsymbol{\theta}}(\boldsymbol{x}) \leq 0$ to a soft constraint in the loss function as $L_{\text{stab}} = \frac{1}{N} \sum_{i=1}^N \left( \mathcal{L}_{\boldsymbol{f}_{\boldsymbol{u}_{\boldsymbol{\phi}}}} V_{\boldsymbol{\theta}}(\boldsymbol{x}_i) + V_{\boldsymbol{\theta}}(\boldsymbol{x}_i) \right)^+$.

**Calculate gradients of $t_k$.** To proceed, we handle the objective function related to the triggering times. Instead of directly training the parameters $\boldsymbol{\phi}, \boldsymbol{\theta}$ based on the direct samples of $V_{\boldsymbol{\theta}}$, $\boldsymbol{u}_{\boldsymbol{\phi}}$ and $\boldsymbol{f}(\boldsymbol{x}, \boldsymbol{u}_{\boldsymbol{\phi}}(\boldsymbol{x}))$ as done in neural certificate-based controllers, we have to numerically solve the controlled ODEs to identify the triggering times. To proceed, we need to calculate the gradients of $t_k$ for optimizing $\frac{1}{t_{k+1} - t_k}$ term in loss function during gradient-based optimization. We employ the neural event ODE method as: $t_{k+1}, \boldsymbol{x}(t_{k+1}) = \text{ODESolveEvent}(\boldsymbol{x}(t_k), \boldsymbol{f}, \boldsymbol{u}_{\boldsymbol{\phi}}, t_k)$, where ODESolveEvent is proposed by (Chen et al., 2020), which introduces root solver and adjoint method (Pontryagin, 2018) to the numerical solver and deduce the gradient $\frac{\partial t_k}{\partial \boldsymbol{\phi}}$ from the implicit function theorem (Krantz & Parks, 2002).

**Reduce computation complexity.** We denote by $t_k(\boldsymbol{x})$ the $k_{\text{th}}$ triggering time from initial value $t_0 = 0$, $\boldsymbol{x}(0) = \boldsymbol{x}$. The computation cost of ODESolveEvent is $\mathcal{O}(M\bar{K}Ld^2)$,

where $M$ is the batch size of the initial value $\{\boldsymbol{x}_i(0)\}_{i=1}^{M}$, $\bar{K} = \frac{1}{M}\sum_{i=1}^{M} K(\boldsymbol{x}_i(0))$, $K(\boldsymbol{x}_i(0)) = \#\{t_k(\boldsymbol{x}_i(0)) : t_k \le T\}$ is the number of triggering times before $T$, and $L$ is the iteration times in the root solver. In this case, the computation cost pivots on the sampled batch and its variance is hard to decrease. In addition, the numerical error in ODE solver accumulates over the triggering time sequence $\{t_k\}$. To mitigate these issues, according to the time invariance property of ODEs, i.e., $t_{k+1}(\boldsymbol{x}(0)) - t_k(\boldsymbol{x}(0)) = t_1(\boldsymbol{x}(t_k))$, we recast the problem of solving $M$ batch trajectories $\{\boldsymbol{x}_i(t_k),\ t_k \le T | \boldsymbol{x}_i(0) \sim q_0(\boldsymbol{x})\}_{i=1}^{M}$ of controlled ODE as solving $MK$ trajectories $\{\boldsymbol{x}_i(t_1),\ t_1 \le T | \boldsymbol{x}_i(0) \sim \tilde{q}_0(\boldsymbol{x})\}_{k=1}^{MK}$ up to $t_1$. Here, $K$ represent the expectation of $\bar{K}$. In practice, we directly treat $MK$ together as a single hyperparameter $M$. Then the triggering times contribute into the loss function as $L_{\text{event}} = \frac{1}{M}\sum_{i=1}^{M} \frac{1}{t_1(\boldsymbol{x}_i(0))}$. Finally, we train the overall parameterized model with the total loss function as follows,

$$L(\boldsymbol{\phi}, \boldsymbol{\theta}) = L_{\text{stab}} + \lambda_1 L_{\text{lip}} + \lambda_2 L_{\text{event}} \qquad (6)$$

The whole training procedure is summarized in Algorithm 1.

*Remark* 3.1. A more reasonable augmented distribution should takes the form as $\tilde{q}_0(\boldsymbol{x}) = \frac{1}{K-1}\sum_{k=0}^{K-1} q_k(\boldsymbol{x})$, where $\boldsymbol{x}_{t_k} \sim q_k(\boldsymbol{x})$ is deduced from the initial distribution $q_0$ and the ODE integration from 0 to $t_k$. Since $t_k$ varies for different initial value and cannot be determined in advance, we fix $\tilde{q}_0 = q_0$ for simplicity.

### 3.2. Monte Carlo Approach

Although the proposed algorithm works efficiently in low dimensional ODEs, the high computation cost and accumulate error caused by the ODE solver affect its performance in higher dimensional tasks. To circumvent this drawback, we propose a Monte Carlo approach for training the Neural ETC. Inspired by the event-triggered scheduling theory in (Tabuada, 2007), we provide the following estimation of minimal inter-event time.

**Theorem 3.2.** *Consider the event-triggered controlled dynamics in Eq.* (1), *if the following assumptions are satisfied:* (i) $\|\boldsymbol{f}(\boldsymbol{x}', \boldsymbol{u}') - \boldsymbol{f}(\boldsymbol{x}, \boldsymbol{u})\| \le l_{\boldsymbol{f}}(\|\boldsymbol{x}' - \boldsymbol{x}\| + \|\boldsymbol{u}' - \boldsymbol{u}\|)$; (ii) $\|\boldsymbol{u}(\boldsymbol{x}') - \boldsymbol{u}(\boldsymbol{x})\| \le l_{\boldsymbol{u}}\|\boldsymbol{x}' - \boldsymbol{x}\|$; (iii) $\mathcal{L}_{\boldsymbol{f}_{\boldsymbol{u}}} V(\boldsymbol{x}, \boldsymbol{u}(\boldsymbol{x} + \boldsymbol{e})) \le -\alpha(\|\boldsymbol{x}\|) + \gamma(\|\boldsymbol{e}\|)$ *for some class-K functions* $\alpha,\ \gamma$ *with* $\alpha^{-1}(\gamma(\|\boldsymbol{e}\|)) \le P\|\boldsymbol{e}\|$. *Then, the minimal inter-event time implicitly defined by event function* $h = \alpha(\|\boldsymbol{x}\|) - \gamma(\|\boldsymbol{e}\|)$ *is lower bounded by* $\tau_h = \frac{1}{l_{\boldsymbol{f}}}\log\frac{P+1}{P+\frac{l_{\boldsymbol{u}}}{1+l_{\boldsymbol{u}}}}$.

The detailed proof is provided in Appendix A.1.2. According to the theorem, the lower bound of minimal inter-event time increases as Lipschitz constants of $\alpha^{-1} \circ \gamma$ and $\boldsymbol{u}$ decrease. Therefore, we can maximize the minimal inter-event time by regularizing these Lipschitz constants. Nonetheless, directly integrating the conditions and results of The-

orem 3.2 into the training process is unrealistic and cumbersome, because the error state $\boldsymbol{e}$ in condition (iii) should depend on $\boldsymbol{x}$ and we cannot determine the sampling distribution of $\boldsymbol{e}$ before training. To solve this problem, we split the inequality in condition (iii) into a sufficient inequality group as

$$\nabla V \cdot (\boldsymbol{f}(\boldsymbol{x}, \boldsymbol{u}(\boldsymbol{x} + \boldsymbol{e})) - \boldsymbol{f}(\boldsymbol{x}, \boldsymbol{u}(\boldsymbol{x}))) \le \gamma(\|\boldsymbol{e}\|) \quad (7)$$

$$\mathcal{L}_{\boldsymbol{f}_{\boldsymbol{u}}} V(\boldsymbol{x}) \le -\alpha(\|\boldsymbol{x}\|) \qquad (8)$$

$$\to \mathcal{L}_{\boldsymbol{f}_{\boldsymbol{u}}} V(\boldsymbol{x}, \boldsymbol{u}(\boldsymbol{x} + \boldsymbol{e})) \le -\alpha(\|\boldsymbol{x}\|) + \gamma(\|\boldsymbol{e}\|)$$

The dependence on $\boldsymbol{x}$ of right term $\gamma$ in Eq. (7) can be omitted when the state space $\mathcal{D}$ is bounded, which occurs in most real-world scenarios. The Eq. (7) implies that the Lipschitz constant of $\gamma$ is related to the Lipschitz constant of $\boldsymbol{u}$. Furthermore, we notice that if we replace the event function in Theorem 3.2 by the following event function with stability guarantee,

$$\tilde{h} = \nabla V \cdot (\boldsymbol{f}(\boldsymbol{x}, \boldsymbol{u}(\boldsymbol{x} + \boldsymbol{e})) - \boldsymbol{f}(\boldsymbol{x}, \boldsymbol{u}(\boldsymbol{x}))) - \alpha(\|\boldsymbol{x}\|), \quad (9)$$

then the inter-event time of these two event functions hold the relation $t_{k+1,h} - t_{k,h} \le t_{k+1,\tilde{h}} - t_{k,\tilde{h}}$ due to Eq. (7). Therefore, the inter-event time of event function $\tilde{h}$ is also lower bounded by $\tau_h$ in Theorem 3.2. We summarize the results in the following theorem.

**Theorem 3.3.** *For the event-triggered controlled dynamics in Eq.* (1) *with event function* $\tilde{h}$ *defined in Eq.* (9), *if the state space* $\mathcal{D}$ *is bounded, the Eqs.* (7),(8) *and the conditions* (i), (ii) *in Theorem 3.2 hold, then the minimal inter-event time is lower bounded by* $\tau_{\tilde{h}} = \frac{1}{l_{\boldsymbol{f}}}\log\frac{cl_{\alpha^{-1}}l_{\boldsymbol{u}}+1}{cl_{\alpha^{-1}}l_{\boldsymbol{u}}+\frac{l_{\boldsymbol{u}}}{1+l_{\boldsymbol{u}}}}$, *here* $l_{\alpha^{-1}}$ *is the Lipschitz constant of* $\alpha^{-1}$, $c$ *is a constant depending on* $V, \boldsymbol{f}, \mathcal{D}$.

The proof is provided in Appendix A.1.3. With this theorem, we come to a Monte Carlo approach for training the Neural ETC framework by directly learning the parameterized functions $V_{\boldsymbol{\theta}}$, $\alpha_{\boldsymbol{\theta}_{\alpha}}$, and control function $\boldsymbol{u}_{\boldsymbol{\phi}}$ simultaneously, as well as regularizing the Lipschitz constants of $\boldsymbol{u}_{\boldsymbol{\phi}}$ and $\alpha_{\boldsymbol{\theta}_{\alpha}}^{-1}$. For constructing neural class-$K$ functions, we adopt the monotonic NNs to construct the candidate class-$\mathcal{K}$ function as

$$\alpha_{\boldsymbol{\theta}_{\alpha}}(x) = \int_0^x q_{\boldsymbol{\theta}_{\alpha}}(s)\mathrm{d}s, \qquad (10)$$

where $q_{\boldsymbol{\theta}_{\alpha}}(\cdot) \ge 0$ is the output of the NNs (Wehenkel & Louppe, 2019). We regularize the inverse of integrand to minimize the Lipschitz constant of $\alpha_{\boldsymbol{\theta}_{\alpha}}^{-1}$. We apply the spectral norm regularization $L_{\text{lip}}$ defined above to minimize the Lipschitz constant of controller $\boldsymbol{u}_{\boldsymbol{\phi}}$. Finally, we train

the overall model with the loss function as follows,

$$\tilde{L}_{\text{stab}} = \frac{1}{N} \sum_{i=1}^{N} \left( \mathcal{L}_{\boldsymbol{f}_{\boldsymbol{u}_{\boldsymbol{\phi}}}} V_{\boldsymbol{\theta}}(\boldsymbol{x}_i) + \alpha_{\boldsymbol{\theta}_{\alpha}}(\boldsymbol{x}_i) \right)^+,$$

$$L_{\alpha^{-1}} = \frac{1}{M_\alpha} \sum_{i=1}^{M_\alpha} \frac{1}{q_{\boldsymbol{\theta}_\alpha}(x_i)}, \tag{11}$$

$$L(\boldsymbol{\phi}, \boldsymbol{\theta}, \boldsymbol{\theta}_\alpha) = \tilde{L}_{\text{stab}} + \lambda_1 L_{\text{lip}} + \lambda_2 L_{\alpha^{-1}}.$$

The specific training procedure of this algorithm, dubbed Neural ETC-MC, is shown in Algorithm 2.

*Remark* 3.4. To obtain a stronger exponential decay rate of $V$, we multiply the term $\alpha(\|\boldsymbol{x}\|)$ in Eq. (9) by $\sigma \in (0, 1)$ in realization. Similarly to Eq. (4), the controlled vector under event $\tilde{h}$ satisfied

$$\nabla V \cdot (\boldsymbol{f}(\boldsymbol{x}, \boldsymbol{u}(\boldsymbol{x} + \boldsymbol{e})) \leq -(1 - \sigma)\alpha(\|\boldsymbol{x}\|). \tag{12}$$

Then the exponential decay rate of $V$ is $1 - \sigma$. The lower bound of inter-event time can be obtained by replacing $l_{\alpha^{-1}}$ with $\sigma^{-1} l_{\alpha^{-1}}$ in Theorem 3.3.

# 4. Theoretical Guarantee for Stability and Optimality

In this section, we provide several theoretical results for rigorously guaranteeing the stability and optimality of our neural controllers. Firstly, we note that the NNs trained on finite samples cannot guarantee the Lyapunov stability condition in the loss function is satisfied in the whole state space with infinite data points. To circumvent this weakness, we introduce the projection operation in the following theorem.

**Theorem 4.1.** *(Stability guarantee) For a candidate controller $\boldsymbol{u}$ and the stable controller space $\mathcal{U}(V) = \{\boldsymbol{u} : \mathcal{L}_{\boldsymbol{f}_{\boldsymbol{u}}} V + V \leq 0\}$, we define the projection operator as,*

$$\pi(\boldsymbol{u}, \mathcal{U}(V)) \triangleq \boldsymbol{u} - \frac{\max(0, \mathcal{L}_{\boldsymbol{f}_{\boldsymbol{u}}} V - V)}{\|\nabla V\|^2} \cdot \nabla V.$$

*If the controller has affine actuator, then we have $\pi(\boldsymbol{u}, \mathcal{U}(V)) \in \mathcal{U}(V)$, the projected controller is Lipschitz continuous over the state space $\mathcal{D}$ if and only if $\mathcal{D}$ is bounded. Furthermore, under the triggering mechanism*

$$\nabla V(\boldsymbol{x}) \cdot [\boldsymbol{f}(\boldsymbol{x}, \boldsymbol{u}(\boldsymbol{x} + \boldsymbol{e})) - \boldsymbol{f}(\boldsymbol{x}, \boldsymbol{u}(\boldsymbol{x}))] - \sigma V(\boldsymbol{x}) = 0,$$
$$\sigma \in (0, 1), \boldsymbol{e} = \boldsymbol{x}(t_k) - \boldsymbol{x}(t), t \in [t_k, t_{k+1})$$

*the controlled system under $\pi(\boldsymbol{u}, \mathcal{U})$ is assured exponential stable with decay rate $1 - \sigma$, and the inter-event time has positive lower bound.*

We provide the proof in Appendix A.1.4. By applying the projection operation to the learned controller $\boldsymbol{u}_{\boldsymbol{\phi}}$ and

potential function $V_{\boldsymbol{\theta}}$, we obtain the theoretical stability guarantee for our approach. Based on the Theorem 4.1 and Theorem 3.2, we could provide necessary condition for the optimal event-triggered control with the largest minimal inter-event time by utilizing the lower bound of the inter-event time and the projection operation.

**Theorem 4.2.** *(Optimality guarantee) Denote the Lipschitz constant of the controller $u$ on state space as $l_u$, then the optimal control with the largest minimal inter-event time satisfies,*

$$\boldsymbol{u} \in \arg\min_{\mathcal{U}(V)} l_{\boldsymbol{u}}. \tag{13}$$

*Furthermore, for any candidate controllers $u$, the optimal condition can be simplified as*

$$\pi(\boldsymbol{u}, \mathcal{U}(V)) \in \arg\min l_{\pi(\boldsymbol{u}, \mathcal{U}(V))}. \tag{14}$$

This theorem is a direct result from the Theorem 4.1 and Theorem 3.2, and the projection operation simplifies the constrained necessary condition in Eq. (13) to the unconstrained condition Eq. (14). We can easily provide optimality guarantee for the neural network controller $\boldsymbol{u}_{\boldsymbol{\phi}}$ and the Lyapunov function $V_{\boldsymbol{\theta}}$ by regularizing the Lipschitz constant of $\pi(\boldsymbol{u}_{\boldsymbol{\phi}}, \mathcal{U}(V_{\boldsymbol{\theta}}))$.

# 5. Experiments and Analysis

In this section, we demonstrate the superiority of the Neural ETCs over existing methods using series of experiments from low dimensional tasks to high dimensional tasks, then we unravel the key factors of Neural ETCs. More details of the experiments can be found in Appendix A.3.

## 5.1. Benchmark Experiments

### Benchmark dynamical systems.

(1) Gene Regulatory Network (GRN) plays a central role in describing the gene expression levels of mRNA and proteins in cell (Davidson & Levin, 2005), here we consider a two-node GRN, $\dot{x}_1 = a_1 \frac{x_1^n}{s^n + x_1^n} + b_1 \frac{s^n}{s^n + x_2^n} - kx_1$, $\dot{x}_2 = a_2 \frac{x_2^n}{s^n + x_2^n} + b_2 \frac{s^n}{s^n + x_1^n} - kx_2$, where the tunable parameters $a_1$, $a_2$, $b_1$ and $b_2$ represent the strengths of auto or mutual regulations. We aim at stabilizing the system from one attractor to another attractor via only tuning $a_1$ in time interval $[0, 20]$.

(2) Lorenz system is a fundamental model in atmospheric science (Lorenz, 1963): $\dot{x} = \sigma(y - x), \dot{y} = \rho x - y - xz, \dot{z} = xy - \beta z$. For this chaotic system, we stabilize its unstable zero solution by an fully actuated controller $\boldsymbol{u} = (\boldsymbol{u}_x, \boldsymbol{u}_y, \boldsymbol{u}_z)$ in time interval $[0, 2]$.

(3) Michaelis–Menten model for subcellular dynamics (Cell) captures the collective behavior of the coupled cells (San-

*Table 1.* Comparison studies of benchmark models and dynamical systems. Best results bolded. Averaged over 5 runs. The dimension of tasks are: GRN (2-D), Lorenz (3-D), Cell (100-D).

| Method | Number of triggers ↓ | | | Minimal inter-event time ↑ | | | MSE under finite triggers ↓ | | |
|---|---|---|---|---|---|---|---|---|---|
| | GRN | Lorenz | Cell | GRN | Lorenz | Cell | GRN | Lorenz | Cell |
| BALSA (Fan et al., 2020) | 12($\pm$4) | 273($\pm$24) | 18($\pm$4) | 0.29($\pm$0.07) | 6e-4($\pm$6e-4) | 3e-3($\pm$1e-3) | 0.05($\pm$0.07) | 7.20($\pm$2.25) | 29.75($\pm$12.72) |
| LQR (Heemels et al., 2012) | 1816($\pm$14) | 2000($\pm$0) | 449($\pm$1) | 6e-3($\pm$3e-3) | 2e-5($\pm$1e-6) | 0.02($\pm$0.02) | 2.19($\pm$0.54) | 53.02($\pm$7.03) | 2e-3($\pm$2e-3) |
| Quad-NLC (Jin et al., 2020) | 1914($\pm$107) | 433($\pm$379) | 551($\pm$220) | 6e-6($\pm$1e-6) | 3e-5($\pm$4e-5) | 4e-6($\pm$10e-7) | 2.29($\pm$0.54) | 7.00($\pm$1.28) | 62.50($\pm$18.44) |
| NLC (Chang et al., 2019) | 23($\pm$2) | 1602($\pm$795) | 15($\pm$13) | 5e-8($\pm$8e-9) | 4e-8($\pm$1e-7) | 6e-6($\pm$1e-5) | 0.20($\pm$0.05) | 40.56($\pm$21.14) | 27.76($\pm$5.52) |
| IRL ETC (Xue et al., 2022) | 131($\pm$28) | 2000($\pm$0) | 370($\pm$14) | 8e-3($\pm$6e-4) | 0.00($\pm$0.00) | 3e-3($\pm$8e-5) | 4.94($\pm$0.77) | 9.76($\pm$2.13) | 38.12($\pm$0.11) |
| Cirtic-Actor ETC (Cheng et al., 2023) | 605($\pm$293) | 69($\pm$8) | 330($\pm$5) | 5e-4($\pm$3e-5) | 2e-3($\pm$8e-4) | 1e-3($\pm$8e-5) | 4.09($\pm$0.81) | 7.22($\pm$2.05) | 38.13($\pm$0.04) |
| ETS (Li & Liu, 2019) | 81($\pm$87) | 1120($\pm$5.62) | 11($\pm$1) | 0.05($\pm$2e-16) | 9e-4($\pm$5e-4) | 0.02($\pm$1e-3) | 0.19($\pm$2.27) | 2.42($\pm$1.02) | 4.96($\pm$7.53) |
| PGDNLC (Yang et al., 2024) | 23($\pm$2) | 340($\pm$246) | 60($\pm$15) | 0.37($\pm$0.15) | 7e-4($\pm$8e-4) | 6e-5($\pm$6e-5) | 0.39($\pm$8e-3) | 11.35($\pm$7.69) | 41.09($\pm$3.36) |
| Neural ETC-PI (ours) | 20($\pm$3) | 20($\pm$4) | 11($\pm$0.00) | 0.26($\pm$0.17) | **0.02($\pm$0.02)** | 0.95($\pm$0.03) | **0.05($\pm$4e-3)** | **0.11($\pm$0.12)** | **5e-8($\pm$1e-9)** |
| Neural ETC-MC (ours) | **4($\pm$4)** | **13($\pm$1)** | **2($\pm$0.00)** | **15.52($\pm$8.54)** | 0.01($\pm$8e-3) | **27.18($\pm$0.60)** | 0.07($\pm$0.03) | 0.29($\pm$0.31) | 1.66($\pm$0.12) |

hedrai et al., 2022): $\dot{x}_i = -Bx_i + \sum_{i=1}^{n} A_{ij} \frac{x_j^2}{1+x_j^2}$. This model has two important equilibrium phase, inactive phase indicating the malignant state and active state indicating the benign state. We consider $n = 100$ and regulate the high dimensional model from the inactive phase to the active phase by only tuning the topology structure $\{A_{ii}\}_{i=1}^n$ in time interval $[0, 30]$.

All these systems have application scenarios and urgently call for event-triggered control with minimal communication burden, we summarize the motivation for selecting the them in Appendix A.3.5.

**Benchmark methods.** We benchmark against the extensively used neural Lyapunov control (NLC) method (Chang et al., 2019), an improvement version of neural Lyapunov control via constructing quadratic Lyapunov function proposed in (Jin et al., 2020), dubbed as Quad-NLC here, a integral reinforcement learning (IRL) based ETC (Xue et al., 2022), a critic-actor neural network based ETC method (Cheng et al., 2023), and two latest SOTA methods ETS (Li & Liu, 2019) and PGDNLC (Yang et al., 2024). We also compare with the classic linear quadratic regulator (LQR) method, BALSA (Fan et al., 2020), an online control policy based on the quadratic programming (QP) solver, and our Neural ETC variants: Neural ETC-PI and Neural ETC-MC. We implement all the control methods with the similar kinds of event functions proposed in Eqs. (3),(9). For a fair comparison, we set the number of hidden units per layer such that all learning models have nearly the same number of total parameters. We provide further details of model selection, hyperparameter selection and experimental configuration in Appendix A.3.

**Results.** Table 1 summarizes the control performance results in terms of the triggering times in the same temporal length, the minimal inter-event time and the mean square error (MSE) between the target state and the controlled trajectories with no larger than 10 triggering events, representing the control performance in limited communication resources. We see that our Neural ETC variants achieve superior performance compared to the other online and offline methods.

For the communication cost, our Neural ETCs need the least number of triggers in the same time interval while have the largest minimal inter-event time compared to other methods, leading to the most optimal scheduling in actual implementation. In addition, the MSE results illustrate our Neural ETCs have the ability in stabilizing the systems at various scales with limited communication resources. We also find the Neural ETC-PI and Neural ETC-MC form the trade off in scheduling and the stabilization performance, we further compare them in the next section.

The results underpin the practicability of the Neural ETCs. Take GRN model for an example, the auto regulation strength $a_1$ can be adjusted externally through the application of repressive or inductive drugs in a typical experimental setting (Wang et al., 2016). In reality, the drugs can only be administered a few times and it takes time for the drug to take effect, requiring the controller should only be updated at several times with large interval. Therefore, while all the benchmark methods successfully regulate the GRN to the target gene expression level in simulation, only the Neural ETC-MC is acceptable.

**Combining online and offline policy.** In the context of event-triggered control, Table 1 demonstrates that the online control method outperforms other offline policies. However, the online policy's computational cost is high due to solving the quadratic programming (QP) problem at each realization time. In contrast, our Neural ETCs achieve superior performance compared to online methods while maintaining the same computation cost as the offline policy during the control process. The event-triggered control employs an event function that continuously assesses whether an event is triggered, effectively acting as an online solver to determine real-time control values. Consequently, we can view event-triggered control as an online realization of the offline policy, inheriting the advantages of both online and offline approaches

## 5.2. Comparison Between Neural ETCs

We further evaluate the strengths and weaknesses of the Neural ETC-PI and Neural ETC-MC. As shown in Table 2, the Neural ETC-MC is more efficient in training process, especially in the high dimensional tasks. Nevertheless, the temporal variance of the controlled trajectories of Neural ETC-PI is far below that of Neural ETC-MC, implying Neural ETC-PI is more robust in the control process. These two algorithms thus are complementary in applications. In addition, the training time of Neural ETC-PI in 2-D GRN and 3-D Lorenz has significant difference, the reason is that the minimal inter-event time of the former is larger than the latter (see Table 1), requiring more time to solve $t_1$.

*Table 2.* Comparison of Neural ETCs (denoted by NETC) in terms of training time and variance of stabilized trajectories.

| Model | Training time ↓ | | Temporal variance ↓ | |
|---|---|---|---|---|
| | NETC-PI | NETC-MC | NETC-PI | NETC-MC |
| GRN | 1230 | **32** | **5e-4** | 7e-3 |
| Lorenz | 503 | **29** | **4e-3** | 0.09 |
| FHN | 4634 | **62** | **3e-15** | 2.78 |

## 5.3. Ablation Study

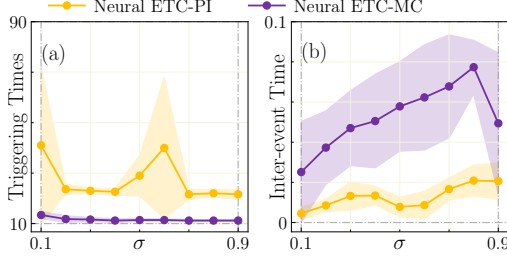

*Figure 3.* The solid lines are obtained through averaging the 5 sampled trajectories, while the shaded areas stand for the variance regions.

The parameter $\sigma$ corresponds to the exponential decay rate of Lyapunov function along controlled trajectory in Eqs. (3),(12). We investigate the influence of $\sigma$ in applying Neural ETC variants to Lorenz dynamic. The results in Fig. 3 suggests the best choice is $\sigma = 0.8$. Then we investigate the influence of weight factor $\lambda_2$ of event loss in Eqs. (6),(11) and summarize the results in Table 3. We find the small $\lambda_2$ leads to poor triggering scheduling because the event loss does not play a leading role in training, the large $\lambda_2$ will break the stabilization performance because the optimization function of event loss is not guaranteed to satisfy the stabilization loss. This phenomenon inspires us to extend the framework to the setting where the parame-

terized controllers are already stabilization controllers in the future work. For reference, in Table 1 Neural ETC-PI is using $\sigma = 0.5$, $\lambda_2 = 0.05$ and Neural ETC-MC is using $\sigma = 0.5$, $\lambda_2 = 0.1$.

*Table 3.* Performance under various event loss weight $\lambda_2$.

| Method | Neural ETC-PI | | | Neural ETC-MC | | |
|---|---|---|---|---|---|---|
| $\lambda_2$ | 0.005 | 0.05 | 0.5 | 0.01 | 0.1 | 1.0 |
| Triggering times ↓ | 114 | 29 | 34 | 37 | **10** | **10** |
| Min Inter-event time ↑ | 0.010 | 0.008 | 0.025 | 0.02 | **0.07** | 0.06 |
| $\langle MSE \rangle_{[1.8,2]}$↓ | **8e-8** | 7e-4 | 0.32 | 3.92 | 0.25 | 0.53 |

## 5.4. Essential Factor of Neural ETC

We investigate the essential factor in the Neural ETC framework that determine the optimization of scheduling. We plot the convexity of $V$ function ($\text{Tr}(\nabla^2 V)$) and the strength of the variation of controller ($\|\nabla u\|$) in the training process, and compare their evolution with triggering times of the corresponding trained controller. Fig. 4 shows that the $\|\nabla u\|$ plays a leading role in minimizing the triggering times as it has strong negative correlation to the triggering times while the convexity of $V$ function does not. We also observe an early convergence phenomenon of the triggering times and $\|\nabla u\|$ simultaneously in Neural ETC-PI from Fig. 4(b),(c).

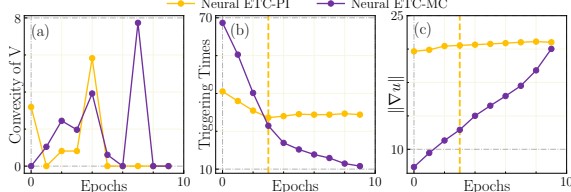

*Figure 4.* (a) Convexity of $V$ is calculated as the trace of the $\nabla^2 V$ on 1000 points in $[-2.5, 2.5]^3$. (b) Triggering times and (c) norm of $\nabla u$ in the training process.

## 6. Related Work

**Neural control with certificate functions.** Previous works in neural control establish the performance guarantee via using the certificate functions, including Lyapunov function for stability (Giesl & Hafstein, 2015; Chang et al., 2019), barrier function for safety (Zhang et al., 2022b; Ames et al., 2016; Taylor et al., 2020; 2019; Taylor & Ames, 2020; Peruffo et al., 2021), and contraction metrics for stability in trajectory tracking (Singh et al., 2021; Tsukamoto et al., 2021). However, all these feedback controllers require impractically high communication cost for updating the controller continuously when deployed on the digital platforms. We solve this challenge in limited communication resources and improve the performance guarantee at the same time.

**Event-triggered control.** The pioneering works (Åström & Bernhardsson, 1999; Åarzén, 1999) highlighted the advantages of event-based control against the periodic implementation in reducing the communication cost. Since then, (Tabuada, 2007) investigates the sufficient conditions for avoiding the Zeno behavior in event-triggered implementations of stabilizing feedback control laws, (Henningsson et al., 2008) extends the event-triggered control to the linear stochastic system and (Heemels et al., 2008) gives the system theory of event-triggered control scheme for perturbed linear systems. Machine learning methods have also been introduced to the ETC settings, (Xue et al., 2022; Cheng et al., 2023) employ the critic-actor RL structure to solve the dynamic Hamilton-Jacobi-Bellman equation under the ETC, (Funk et al., 2021) cultivates a model-free hierarchical RL method to optimize both the control and communication policies for discrete dynamics, and (Baumann et al., 2018) applies deep RL to ETC in the nonlinear systems. All the previous works focus on the stabilization analysis of the controlled systems, the existence of the minimal inter-event time (and hence avoids the Zeno behavior), and directly introducing machine learning methods to ETC. To our knowledge, we are the first to study the optimization scheduling problem of ETC in the continuous dynamics.

## 7. Scope and Limitations

**ODE solver.** The use of the fixed step ODE solvers in finding the triggering times in the training process is less optimal than the adaptive ODE solver. One can still improve the performance of the framework by applying the adaptive solvers with higher accuracy tolerance with a stronger computing platform. However, in practice the performance of the Neural ETC did not decrease substantially when using adaptive solvers. In addition, the employ of ODE solvers in the Neural ETC-PI may not always work, especially for systems described by stiff equations, stiff-based ODE solvers can be introduced to mitigate this issue (Kim et al., 2021).

**Neural ETC for SDEs.** Although the current Neural ETC framework works efficiently in ODEs, many real-world scenarios affected by the noise are described by stochastic differential equations (SDEs). The major challenge for establishing the Neural ETC framework for SDEs ensues from the stochasticity of the triggering time. Specifically, the triggering time in SDEs, $t_1 = \inf_{t \geq 0}\{t : h(\boldsymbol{x}(t)) = 0\}$ initiated from any fixed $\boldsymbol{x}(0)$ with $h(\boldsymbol{x}(0)) < 0$, is a stopping time. Therefore, $t_1$ is a random variable and can take different values in different sample paths. In this case, none of the existing methods can find $t_1$ for SDEs as a counterpart of `ODESolveEvent` for ODEs.

## 8. Conclusion

This work focuses on a new connection of machine learning and control field in the context of learning event-triggered stabilization control with optimal scheduling. In contrast to the existing learning control methods, the learned event-triggered control, named Neural ETC, only updates the control value in very few times when an event is triggered. As a consequence, our Neural ETC can be deployed on the actual platform where the communication cost for updating the control value is limited (e.g. tuning the protein regulation strength in cell via drugs). The superiority of the Neural ETC over the existing methods is demonstrated through a series of representative dynamical systems.

## Acknowledgements

L. Yang is supported by the China Scholarship Council (No. 202406100251). Q. Zhu is supported by the China Postdoctoral Science Foundation (Grant No. 2022M720817), by the Shanghai Postdoctoral Excellence Program (Grant No. 2021091), and by the STCSM (Grants No. 21511100200, No. 22ZR1407300, and No. 23YF1402500). W. Lin is supported NSFC (Grant No. 11925103), the IPSMEC (Grant No. 2023ZKZD04), and the STCSM (Grants No. 22JC1401402, No. 22JC1402500, and No. 2021SHZDZX0103).

## Impact Statement

This paper presents work whose goal is to advance the field of Machine Learning. There are many potential societal consequences of our work, none which we feel must be specifically highlighted here.

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

# A. Appendix

## A.1. Proofs and Derivations

In this section, we introduce some basic notations and then provide the proofs of the theoretical results.

### A.1.1. NOTATIONS

**Notations.** Throughout the paper, we employ the following notation. Let $\langle x, y \rangle$ be the inner product of vectors $x, y \in \mathbb{R}^d$. For a second continuous function $f(x) : \mathbb{R}^d \to \mathbb{R}$, let $\nabla f$ denote the gradient of $f(x)$, that is, $\nabla^2 f$ denote the Hessian matrix of $f$. For the two sets $A, B$, let $A \subset B$ denote that $A$ is covered in $B$. Denote by $\log$ the base $e$ logarithmic function. Denote by $\| \cdot \|$ the $L^2$-norm for any given vector in $\mathbb{R}^d$. Denote by $| \cdot |$ the absolute value of a scalar number or the modulus length of a complex number. For $A = (a_{ij})$, a matrix of dimension $d \times r$, denote by $\|A\|_F^2 = \sum_{i=1}^{d} \sum_{j=1}^{r} a_{ij}^2$ the Frobenius norm.

### A.1.2. PROOF OF THEOREM 3.2

**Theorem A.1.** *Consider the event-triggered controlled dynamics in Eq. (1), if the following assumptions are satisfied:* (i) $\|f(x', u') - f(x, u)\| \le l_f (\|x' - x\| + \|u' - u\|)$; (ii) $\|u(x') - u(x)\| \le l_u \|x' - x\|$; (iii) $\mathcal{L}_{f_u} V(x, u(x + e)) \le -\alpha(\|x\|) + \gamma(\|e\|)$ *for some class-K functions* $\alpha$, $\gamma$ *with* $\alpha^{-1}(\gamma(\|e\|)) \le P\|e\|$. *Then, the minimal inter-event time implicitly defined by event function* $h = \alpha(\|x\|) - \gamma(\|e\|)$ *is lower bounded by* $\tau_h = \frac{1}{l_f} \log \frac{P+1}{P + \frac{l_f l_u}{l_f(1+l_u)}}$.

From the condition (iii) and the definition of the event function, we have the triggering time happens after $P\|e\| = \|x\|$. Therefore, the inter-event time is lower bounded by the minimal inter-event time defined by the event function $\tilde{h} = P(\|e\|) - \|x\|$. Now we come to deduce the estimation of the inter-event time of $\tilde{h}$, i.e., the time from $\|e\| = 0$ to $\|e\| = \frac{1}{P}\|x\|$. Consider the dynamic of $\frac{\|e\|}{\|x\|}$, we have

$$
\begin{aligned}
\frac{\mathrm{d}}{\mathrm{d}t} \frac{\|e\|}{\|x\|} &= \frac{\mathrm{d}}{\mathrm{d}t} \frac{(e^\top e)^{1/2}}{(x^\top x)^{1/2}} \\
&= \frac{\frac{1}{2}(e^\top e)^{-1/2} 2 e^\top \dot{e} (x^\top x)^{1/2} - \frac{1}{2}(x^\top x)^{-1/2} 2 x^\top \dot{x} (e^\top e)^{1/2}}{x^\top x} \\
&= \frac{e^\top \dot{e}}{\|e\| \|x\|} - \frac{x^\top \dot{x}}{\|x\| \|x\|} \frac{\|e\|}{\|x\|} \\
&= -\frac{e^\top \dot{x}}{\|e\| \|x\|} - \frac{x^\top \dot{x}}{\|x\| \|x\|} \frac{\|e\|}{\|x\|} \\
&\le \frac{\|e\| \|\dot{x}\|}{\|e\| \|x\|} + \frac{\|x\| \|\dot{x}\|}{\|x\| \|x\|} \frac{\|e\|}{\|x\|} \\
&= \frac{\|\dot{x}\|}{\|x\|} \left( 1 + \frac{\|e\|}{\|x\|} \right) \\
&= \frac{\|f(x, u(x + e))\|}{\|x\|} \left( 1 + \frac{\|e\|}{\|x\|} \right) \\
&\le \frac{l_f \|x\| + l_f l_u (\|x\| + \|e\|)}{\|x\|} \left( 1 + \frac{\|e\|}{\|x\|} \right) \\
&= \left( l_f(1 + l_u) + l_f l_u \frac{\|e\|}{\|x\|} \right) \left( 1 + \frac{\|e\|}{\|x\|} \right).
\end{aligned}
$$

By denoting $z = \frac{\|e\|}{\|x\|}$, we have the triggering time of $\tilde{h}$ happens after the variable $z$ increases from $0$ to $\frac{1}{P}$. The dynamic of $z$ is

$$
\begin{aligned}
\dot{z} &= (l_f(1 + l_u) + l_f l_u z)(1 + z) \\
z_0 &= 0, \\
z_T &= \frac{1}{P}.
\end{aligned}
$$

We have

$$\frac{\mathrm{d}z}{(1+az)(1+y)} = b\mathrm{d}t,$$

where $a = \frac{l_f l_u}{l_f(1+l_u)}$, $b = l_f(1+l_u)$. Then we have

$$\begin{aligned}
\frac{\mathrm{d}z}{(1+az)(1+z)} &= \frac{a}{a-1}\left(\frac{1}{1+az} - \frac{1}{a(1+z)}\right)\mathrm{d}z \\
&= \frac{1}{a-1}\left(\mathrm{d}\log(1+az) - \mathrm{d}\log(1+z)\right) \\
&= b\mathrm{d}t
\end{aligned}$$

By integrating the above equation, we have

$$\begin{aligned}
\frac{1}{a-1}\left(\log(1+\frac{a}{P}) - \log(1+\frac{1}{P})\right) &= bT \\
\rightarrow T &= \frac{1}{b(a-1)}\log\left(\frac{1+\frac{a}{P}}{1+\frac{1}{P}}\right) \\
&= \frac{1}{b(1-a)}\log\left(\frac{1+\frac{1}{P}}{1+\frac{a}{P}}\right) \\
&= \frac{1}{l_f}\log\frac{P+1}{P+\frac{l_f l_u}{l_f(1+l_u)}},
\end{aligned}$$

which completes the proof.

### A.1.3. PROOF OF THEOREM 3.3

**Theorem A.2.** *For the event-triggered controlled dynamics in Eq. (1) with event function $\tilde{h}$ defined in Eq. (9), if the state space $\mathcal{D}$ is bounded, the Eqs. (7),(8) and the conditions (i), (ii) in Theorem 3.2 hold, then the minimal inter-event time is lower bounded by $\tau_{\tilde{h}} = \frac{1}{l_f}\log\frac{cl_{\alpha^{-1}}l_u+1}{cl_{\alpha^{-1}}l_u+\frac{l_f l_u}{l_f(1+l_u)}}$, here $l_{\alpha^{-1}}$ is the Lipschitz constant of $\alpha^{-1}$.*

From the Eqs. (7), we know that the triggering time defined by $\tilde{h}$ in Eq. 9 is larger than that defined by $h$ in Theorem 3.2. Notice in Theorem 3.2 $P$ is a tight upper bound Lipschitz constant of $\alpha^{-1} \circ \gamma$. Since the state space $\mathcal{D}$ is bounded, from Eq. 7, if we set $\gamma$ as the tight estimation of $\nabla V \cdot (f(x, u(x+e)) - f(x, u(x)))$, the Lipschitz constant of $\gamma$ can be bounded by

$$\max_{x \in \mathcal{D}} \|\nabla V(x)\| l_f l_u.$$

Then we get

$$\mathrm{Lip}(\alpha^{-1} \circ \gamma) \leq \max_{x \in \mathcal{D}} \|\nabla V(x)\| l_f l_u l_{\alpha^{-1}}.$$

By denoting $c = \max_{x \in \mathcal{D}} \|\nabla V(x)\| l_f$ and replace $P$ with $cl_{\alpha^{-1}}l_u$ in Theorem 3.2, we obtain the final estimation of $\tau_{\tilde{h}}$.

A.1.4. PROOF OF THEOREM 4.1

**Theorem A.3.** *(Stability guarantee) For a candidate controller $\boldsymbol{u}$ and the stable controller space $\mathcal{U}(V) = \{\boldsymbol{u} : \mathcal{L}_{\boldsymbol{f}_{\boldsymbol{u}}}V + V \leq 0\}$, we define the projection operator as,*

$$\pi(\boldsymbol{u}, \mathcal{U}(V)) \triangleq \boldsymbol{u} - \frac{\max(0, \mathcal{L}_{\boldsymbol{f}_{\boldsymbol{u}}}V - V)}{\|\nabla V\|^2} \cdot \nabla V.$$

*If the controller has affine actuator, then we have $\pi(\boldsymbol{u}, \mathcal{U}(V)) \in \mathcal{U}(V)$, the projected controller is Lipschitz continuous over the state space $\mathcal{D}$ if and only if $\mathcal{D}$ is bounded. Furthermore, under the triggering mechanism*

$$\nabla V(\boldsymbol{x}) \cdot [\boldsymbol{f}(\boldsymbol{x}, \boldsymbol{u}(\boldsymbol{x} + \boldsymbol{e})) - \boldsymbol{f}(\boldsymbol{x}, \boldsymbol{u}(\boldsymbol{x}))] - \sigma V(\boldsymbol{x}) = 0,$$
$$\sigma \in (0, 1), \boldsymbol{e} = \boldsymbol{x}(t_k) - \boldsymbol{x}(t), t \in [t_k, t_{k+1})$$

*the controlled system under $\pi(\boldsymbol{u}, \mathcal{U})$ is assured exponential stable with decay rate $1 - \sigma$, and the inter-event time has positive lower bound.*

**Proof.** To begin with, we check the inequality constraint in $\mathcal{U}(V)$ is satisfied by the projection element, that is

$$\mathcal{L}_{\boldsymbol{f}_{\boldsymbol{u}}}V\big|_{\boldsymbol{u}=\pi(\boldsymbol{u},\mathcal{U}(V))} \leq -V.$$

Since the controller has affine actuator, from the definition of the Lie derivative operator, we have

$$
\begin{aligned}
\mathcal{L}_{\boldsymbol{f}_{\boldsymbol{u}}}V\big|_{\boldsymbol{u}=\pi(\boldsymbol{u},\mathcal{U}(V))} &= \nabla V \cdot (\boldsymbol{f} + \boldsymbol{u} - \frac{\max(0, \mathcal{L}_{\boldsymbol{u}}V + V)}{\|\nabla V\|^2} \cdot \nabla V) \\
&= \nabla V \cdot (\boldsymbol{f} + \boldsymbol{u}) - \nabla V \cdot \frac{\max(0, \mathcal{L}_{\boldsymbol{u}}V + V)}{\|\nabla V\|^2} \cdot \nabla V \\
&= \mathcal{L}_{\boldsymbol{u}}V - \max(0, \mathcal{L}_{\boldsymbol{u}}V + V) \leq -V.
\end{aligned}
$$

Next, we show the equivalent condition of the Lipschitz continuity of projection element. Notice that $\boldsymbol{u} \in \mathrm{Lip}(\mathcal{D})$, then we have

$$\hat{\pi}(\boldsymbol{u}, \mathcal{U}(V)) \in \mathrm{Lip}(\mathcal{D}) \iff \frac{\max(0, \mathcal{L}_{\boldsymbol{u}}V + V)}{\|\nabla V\|^2} \cdot \nabla V \in \mathrm{Lip}(\mathcal{D}).$$

Since $\|\frac{\nabla V}{\|\nabla V\|}\|$ is a continuous unit vector, and naturally is Lipschitz continuous, we only need to consider the remaining term $\frac{\max(0, \mathcal{L}_{\boldsymbol{u}}V + V)}{\|\nabla V\|}$. According to the definition, all the functions occured in this term are continuous, so we only need to bound this term to obtain the global Lipschitz continuity, that is

$$\frac{\max(0, \mathcal{L}_{\boldsymbol{u}}V + V)}{\|\nabla V\|} \in \mathrm{Lip}(\mathcal{D}) \iff \sup_{\boldsymbol{x} \in \mathcal{D}} \frac{\max(0, \mathcal{L}_{\boldsymbol{u}}V + V)}{\|\nabla V\|} < +\infty.$$

When $\mathcal{L}_{\boldsymbol{u}}V \leq -V$, obviously we have $\max(0, \mathcal{L}_{\boldsymbol{u}}V + V) = 0 < +\infty$, otherwise, since $V \geq \varepsilon \|\boldsymbol{x}\|^p$, we have

$$\mathcal{L}_{\boldsymbol{u}}V + V \geq \mathcal{L}_{\boldsymbol{u}}V + \varepsilon\|\boldsymbol{x}\|^p \approx \mathcal{O}(\|\boldsymbol{x}\|^p) \to \infty(\|\boldsymbol{x}\| \to \infty).$$

Thus, we have

$$\sup_{\boldsymbol{x} \in \mathcal{D}} \frac{\max(0, \mathcal{L}_{\boldsymbol{u}}V + V)}{\|\nabla V\|} < +\infty \iff \sup_{\boldsymbol{x} \in \mathcal{D}} \|\boldsymbol{x}\| < +\infty.$$

The positive lower bound of the inter-event time comes from the Theorem 3.2. We complete the proof.

## A.2. Algorithms

In this section, we provide the algorithms of Neural ETC-PI (1) and Neural ETC-MC (2). Firstly, we supplement the warm up stage for path integral algorithm to accelerate the convergence of training process.

**Warm up.**  At the beginning of the training process, the stability constraint is not satisfied, which leads to the solution $t_1$ of the event function $h_{\boldsymbol{\theta},\boldsymbol{\phi}}$ does not exist. To ensure the training process can proceed smoothly, we pre-train the parameterized model with

$$\tilde{L}(\boldsymbol{\phi}, \boldsymbol{\theta}, \{c_i\}) = L_{\text{stab}} + \lambda_1 L_{\text{lip}}. \tag{15}$$

---

**Algorithm 1** Neural ETC-PI: Path Integral Algorithm

---

1: **hyperparameters:**
    $N, M$      ▷ Sample size and batch size
    $\beta, m$      ▷ Learning rate and max iterations
    $\mu(\mathcal{D}), \lambda_1, \lambda_2$      ▷ Data distrituion, weight factors
2: **initialize** $\boldsymbol{w} = (\boldsymbol{\phi}, \boldsymbol{\theta})$      ▷ From Eq. (5)
3: **generate dataset** $\mathcal{D}_N = \{\boldsymbol{x}_i\}_{i=1}^N \sim \mu(\mathcal{D})$
4: **for** $r = 1 : m$ **do**
5:     $\boldsymbol{w} \leftarrow \boldsymbol{w} - \beta \nabla_{\boldsymbol{w}} \tilde{L}(\boldsymbol{w})$      ▷ Warm up in Eq. (15)
6: **end for**
7: **for** $r = 1 : m$ **do**
8:     $\{\boldsymbol{x}_i(0)\}_{i=1}^M \sim \mathcal{D}_N$      ▷ Sample batch data
9:     $t_{i,1}, \boldsymbol{x}_i(t_{i,1}) = \texttt{ODESolveEvent}(\boldsymbol{x}_i(0), \boldsymbol{f}, \boldsymbol{u}_{\boldsymbol{\phi}}, 0)$
10:     $\boldsymbol{w} \leftarrow \boldsymbol{w} - \beta \nabla_{\boldsymbol{w}} L(\boldsymbol{w})$      ▷ From Eq. (6)
11: **end for**
12: **return** $\boldsymbol{u}_{\boldsymbol{\phi}}, V_{\boldsymbol{\theta}}$

---

**Algorithm 2** Neural ETC-MC: Monte Carlo Algorithm

---

1: **hyperparameters:**
    $N, M_\alpha, \lambda_1, \lambda_2$      ▷ Sample sizes and weight factors
    $\beta, m$      ▷ Learning rate and max iterations
    $\mu(\mathcal{D}), \mu(\mathcal{X})$      ▷ Distributions of state and error
2: **initialize** $\boldsymbol{w} = (\boldsymbol{\phi}, \boldsymbol{\theta}, \boldsymbol{\theta}_\alpha)$      ▷ Eqs. (5),(10)
3: **generate dataset** $\{\boldsymbol{x}_i\}_{i=1}^N \times \{x_i\}_{i=1}^{M_\alpha} \sim \mu(\mathcal{D}) \times \mu(\mathcal{X})$
4: **for** $r = 1 : m$ **do**
5:     $\boldsymbol{w} \leftarrow \boldsymbol{w} - \beta \nabla_{\boldsymbol{w}} L(\boldsymbol{w})$      ▷ From Eq. (11)
6: **end for**
7: **return** $\boldsymbol{u}_{\boldsymbol{\phi}}, V_{\boldsymbol{\theta}}, \alpha_{\boldsymbol{\theta}_\alpha}$

---

### A.3. Experimental Configurations

In this section, we provide the detailed descriptions for the experimental configurations of the benchmark dynamical systems and control methods in the main text. We implement the code on a single i7-10870 CPU with 16GB memory, and we train all the parameters with Adam optimizer.

#### A.3.1. NEURAL NETWORK STRUCTURES

- For constructing the potential function $V$, we utilize the ICNN as (Amos et al., 2017):

$$
\begin{aligned}
\boldsymbol{z}_1 &= \sigma(W_0\boldsymbol{x} + b_0), \\
\boldsymbol{z}_{i+1} &= \sigma(U_i\boldsymbol{z}_i + W_i\boldsymbol{x} + b_i), \; i = 1, \cdots, k-1, \\
p(\boldsymbol{x}) &\equiv \boldsymbol{z}_k, \\
V(\boldsymbol{x}) &= \sigma(p(\boldsymbol{x}) - p(\boldsymbol{0})) + \varepsilon\|\boldsymbol{x}\|^2,
\end{aligned}
$$

where $\sigma$ is the smoothed **ReLU** function as defined in the main text, $W_i \in \mathbb{R}^{h_i \times d}$, $U_i \in (\mathbb{R}_+ \cup \{0\})^{h_i \times h_{i-1}}$, $\boldsymbol{x} \in \mathbb{R}^d$, and, for simplicity, this ICNN function is denoted by $\mathrm{ICNN}(h_0, h_1, \cdots, h_{k-1})$. We set $\varepsilon = 1\text{e-}3$ as default value for all the experiments;

- The class-$\mathcal{K}$ function $\alpha$ is constructed as:

$$
\begin{aligned}
\boldsymbol{q}_1 &= \mathrm{ReLU}(W_0 s + b_0), \\
\boldsymbol{q}_{i+1} &= \mathrm{ReLU}(W_i\boldsymbol{q}_i + b_i), i = 1, \cdots, k-2, \\
\boldsymbol{q}_k &= \mathrm{ELU}(W_{k-1}\boldsymbol{q}_{k-1} + b_{k-1}), \\
\alpha(x) &= \int_0^x q_k(s)\mathrm{d}s
\end{aligned}
$$

where $W_i \in \mathbb{R}^{h_{i+1} \times h_i}$, and this class-$\mathcal{K}$ function is denoted by $\mathcal{K}(h_0, h_1, \cdots, h_k)$;

- The neural control function (nonlinear version) is constructed as:

$$
\begin{aligned}
\boldsymbol{z}_1 &= \mathcal{F}(\texttt{SpectralNorm}(W_0\boldsymbol{x} + b_0)), \\
\boldsymbol{z}_{i+1} &= \mathcal{F}(\texttt{SpectralNorm}((W_i\boldsymbol{z}_i + b_i)), \; i = 1, \cdots, k-1, \\
\mathbf{NN}(\boldsymbol{x}) &\equiv \texttt{SpectralNorm}(W_k\boldsymbol{z}_k), \\
\boldsymbol{u}(\boldsymbol{x}) &= \mathrm{diag}(\boldsymbol{x} - \boldsymbol{x}^*)\mathbf{NN}(\boldsymbol{x}) \text{ or } \mathbf{NN}(\boldsymbol{x}) - \mathbf{NN}(\boldsymbol{x}^*),
\end{aligned}
$$

where $\mathcal{F}(\cdot)$ is the activation function, $\texttt{SpectralNorm}$ is the spectral norm function from (Yoshida & Miyato, 2017), $W_i \in \mathbb{R}^{h_{i+1} \times h_i}$, and this control function is denoted by $\mathrm{Control}(h_0, h_1, \cdots, h_{k+1})$. Since we deploy the $\texttt{SpectralNorm}$ package in our algorithm, the weight factor $\lambda_1$ for Lipschitz constant of $\boldsymbol{u}$ is automatically set as the default value in this package and we do not tune it in our experiments due to its good performance.

- The standard neural network is constructed as:

$$
\begin{aligned}
\boldsymbol{z}_1 &= \mathcal{F}(W_0\boldsymbol{x} + b_0), \\
\boldsymbol{z}_{i+1} &= \mathcal{F}(W_i\boldsymbol{z}_i + b_i), \; i = 1, \cdots, k-1, \\
\mathbf{NN}(\boldsymbol{x}) &\equiv W_k\boldsymbol{z}_k,
\end{aligned}
$$

where $\mathcal{F}(\cdot)$ is the activation function, and this standard function is denoted by $\mathrm{MLP}(h_0, h_1, \cdots, h_{k+1})$

#### A.3.2. GENE REGULATORY NETWORK

Here we model the controlled gene regulatory network (GRN) as

$$
\begin{aligned}
\dot{x}_1 &= a_1 \frac{x_1^n}{s^n + x_1^n} + b_1 \frac{s^n}{s^n + x_2^n} - kx_1 + u\frac{x_1^n}{s^n + x_1^n}, \\
\dot{x}_2 &= a_2 \frac{x_2^n}{s^n + x_2^n} + b_2 \frac{s^n}{s^n + x_1^n} - kx_2,
\end{aligned}
$$

where the under-actuated control $u$ only acts on the protein regulation strength $a_1$. We specify $a_1 = a_2 = 1$, $b_1 = b_2 = 0.2$, $n = 2$, $k = 1.1$, $s = 0.5$. The two attractors of the original model is

$$\boldsymbol{P}_1 : (x_1^*, x_2^*) = (0.62562059, 0.62562059),$$
$$\boldsymbol{P}_2 : (x_1^0, x_2^0) = (0.0582738, 0.85801853).$$

We aims at stabilize the attractor $\boldsymbol{P}_2$ with low protein concentration to $\boldsymbol{P}_1$ with high protein expression level. We slightly modify the neural networks s.t. $V(\boldsymbol{P}_1) = 0$, $u(\boldsymbol{P}_1) = 0$, e.g. $V = V(\boldsymbol{x}) - V(\boldsymbol{P}_1)$, $u = u(\boldsymbol{x}) - u(\boldsymbol{P}_1)$. Since these two attractors are close in the Euclidean space, it hard for algorithms to identify them from states with numerical error. To address this issue, we rescale the original system as $\tilde{x}_1 = 10x_1$, $\tilde{x}_2 = 10x_2$ to enlarge the attractors. For training controller $\boldsymbol{u}$, we uniformly sample 1000 data from the state region $[-10, 10]$. We test the performance under different learning rate $\mathrm{lr} \in \{0.01, 0.03, 0.05\}$ and pick the best one, the considered control methods are set as following,

**Neural ETC-PI.** We parameterize $V(\boldsymbol{x})$ as ICNN$(2, 10, 10, 1)$, $\boldsymbol{u}(\boldsymbol{x})$ as Control$(2, 20, 20, 1)$ with $\mathcal{F} = \mathrm{ReLU}$. We set the iterations for warm up as 500, the iterations and batch size for calculating the triggering times as 50 and 10, the learning rate as $\mathrm{lr} = 0.01$, the weight factor for event loss as $\lambda_2 = \frac{10}{1000}$.

**Neural ETC-MC.** We parameterize $V(\boldsymbol{x})$ as ICNN$(2, 20, 1)$, $\boldsymbol{u}(\boldsymbol{x})$ as Control$(2, 20, 20, 1)$. We set the iterations as $500 + 50$, the learning rate as $\mathrm{lr} = 0.05$, the weight factor for event loss as $\lambda_2 = 0.1$.

**NLC.** We parameterize $V(\boldsymbol{x})$ as MLP$(2, 20, 20, 1)$, $\boldsymbol{u}(\boldsymbol{x})$ as MLP$(2, 20, 20, 1)$. We set the iterations as $500 + 50$, the learning rate as $\mathrm{lr} = 0.05$, the loss function is

$$L = \frac{1}{N} \sum_{i=1}^{N} \left[ \left( \mathcal{L}_{\boldsymbol{f}_{\boldsymbol{u}_\phi}} V_\theta(\boldsymbol{x}_i) \right)^+ + (V_\theta(\boldsymbol{x}_i))^+ \right] + V_\theta(P_1)^2$$

**Quad-NLC.** We parameterize $V(\boldsymbol{x})$ as $(\boldsymbol{x} - \boldsymbol{P}_1)^\top \mathrm{MLP}(2, 20, 2)^\top \mathrm{MLP}(2, 20, 2)(\boldsymbol{x} - \boldsymbol{P}_1)$, $\boldsymbol{u}(\boldsymbol{x})$ as MLP$(2, 20, 20, 1)$. We set the iterations as $500 + 50$, the learning rate as $\mathrm{lr} = 0.05$, the loss function is

$$L = \frac{1}{N} \sum_{i=1}^{N} \left( \mathcal{L}_{\boldsymbol{f}_{\boldsymbol{u}_\phi}} V_\theta(\boldsymbol{x}_i) + V_\theta(\boldsymbol{x}_i) \right)^+ + V_\theta(P_1)^2.$$

**BALSA.** For this QP based method, we set the object function as

$$\min_{u, d_1, d_2} \frac{1}{2} \|u\|^2 + p_1 d_1^2,$$
$$\mathrm{s.t.} \mathcal{L}_{\boldsymbol{f}_u} V - V \le d_1,$$

where $d_1$ is the relaxation number. We choose $V = \frac{1}{2} \|\boldsymbol{x} - \boldsymbol{P}_1\|^2$, $p_1 = 50$. We solve this problem with the QP solver in `cvxopt` in Python package.

**LQR.** We linearize the controlled dynamic near the target $\boldsymbol{P}_1$ as

$$\dot{\boldsymbol{x}} = \boldsymbol{A}(\boldsymbol{x} - \boldsymbol{P}_1) + \boldsymbol{B}u,$$

$$\boldsymbol{A} = \begin{pmatrix} a_1 \dfrac{n(x_1^*)^{n-1}}{(s^n + (x_1^*)^n)^2} - k & -b_1 \dfrac{n(x_2^*)^{n-1}}{(s^n + (x_2^*)^n)^2} \\ -a_2 \dfrac{n(x_1^*)^{n-1}}{(s^n + (x_1^*)^n)^2} & b_2 \dfrac{n(x_2^*)^{n-1}}{(s^n + (x_2^*)^n)^2} - k \end{pmatrix},$$

$$\boldsymbol{B} = \begin{pmatrix} \dfrac{(x_1^*)^n}{(s^n + (x_1^*)^n)^2} - k \\ 0 \end{pmatrix}.$$

We set the cost matrix in LQR as

$$Q = \begin{pmatrix} 10 & 0 \\ 0 & 10 \end{pmatrix},$$
$$R = (0.1)$$

and solve the problem via `lqr` method in Matlab. The obtained Riccati solution $S$ forms the Lyapunov function $V = \frac{1}{2}(x - P_1)^\top S(x - P_1)$, the controller is $u = -K(x - P_1)$ where $K \in \mathbb{R}^{1\times 2}$ is returned by the lqr solver. The Lie derivative of the Lyapunov function is $-(x - P_1)^\top Q_1(x - P_1)$ with $Q_1 = Q + K^\top RK$.

**Critic-Actor ETC.** According to the implementation setting in (Cheng et al., 2023), we consider the following event-triggered controller parametrized by the critic neural network $W_c$ and the actor neural network $W_a$,

$$\dot{x} = f(x) + g(x)u(x),$$
$$V^*(x) = \min_u \int_0^T (x^\top Q x + u^\top R u)\, \mathrm{d}t,$$
$$V^*(x) = x^\top W_c x,$$
$$u^*(x) = W_a x,$$
$$e_a = W_a x + \frac{1}{2} g^\top \nabla V^*(x),$$
$$K_a = \frac{1}{2} e_a^\top e_a,$$
$$\dot{W}_a = -\frac{\partial K_a}{\partial W_a},$$
$$e_c = \nabla V^*(x) \cdot [f(x) + g(x)u(x)] + x^\top Q x + u^\top R u,$$
$$K_c = \frac{1}{2} e_c^\top e_c,$$
$$\dot{W}_c = -\frac{\partial K_c}{\partial W_c},$$

here $f$ is the original dynamics described above, $g$ is the actuator taking the form,

$$g = \begin{pmatrix} \dfrac{x_1^n}{(s^n + x_1^n)^2} \\ 0 \end{pmatrix}.$$

The cost matrix $Q$, $R$ are the same as that in LQR. In the event-triggered mode, the weights of critic and actor NN, $W_c$ and $W_a$, obeying the evolution dynamics as follows,

$$\dot{W}_a = 0,\ t \in [t_k, t_{k+1}),$$
$$W_a^+ = W_a - \alpha_a \frac{\partial K_a}{\partial W_a}, t = t_{k+1},$$
$$\dot{W}_c = 0,\ t \in [t_k, t_{k+1}),$$
$$W_c^+ = W_c - \alpha_c \frac{\partial K_c}{\partial W_c}, t = t_{k+1},$$

where $\alpha_a$ and $\alpha_c$ are the learning rates of the critic and actor NNs, respectively. For the initial value of $W_c$ and $W_a$, we employ the solutions from the above LQR solver as $W_c = S$, $W_a = -K$. We set the learning rate as $\alpha_c = \alpha_a = 1e-2$, the event function is set as $h = |e| - e_{\text{thres}}$, $e_{\text{thres}} = 0.2$ according to (Cheng et al., 2023), here $e = (e_{x_1}, e_{x_2})$ are the variables of error dynamics. The event function here is different to our proposed stability guaranteed function because of the lack of Lyapunov function in this method, we note that $V^*$ is only an auxiliary function used to find the dynamics of $W_c$, $W_a$ and cannot be verified as a Lyapunov function. We have tuned the hyperparameters $\alpha_c, \alpha_a \in \{5e-4, 1e-3, 1e-2,, 1e-1\}$, $e_{\text{thres}} \in \{0.1, 0.2, 0.3, 0.4, 0.5\}$ and fix the parameters with the best performance.

**IRL ETC.** Similarly to the Critic-Actor ETC, (Xue et al., 2022) transformed the optimization control problem to a RL problem via abstracting the Hamilton-Jacobi-Bellman equation as the value function and approximating the optimal value function based on a preset basis activation function. Specifically, we consider the control problem parametrized by the critic neural network $W$ as follows,

$$\dot{\boldsymbol{x}} = \boldsymbol{f}(\boldsymbol{x}) + \boldsymbol{g}(\boldsymbol{x})\boldsymbol{u}(\boldsymbol{x}),$$

$$V^*(\boldsymbol{x}) = \min_{\boldsymbol{u}} \int_0^T \left(\boldsymbol{x}^\top \boldsymbol{Q}\boldsymbol{x} + \boldsymbol{u}^\top \boldsymbol{R}\boldsymbol{u}\right) \mathrm{d}t,$$

$$V^*(\boldsymbol{x}) = \boldsymbol{x}^\top \boldsymbol{W}\boldsymbol{x},$$

$$u^*(\boldsymbol{x}) = \eta\sigma\left(-\frac{1}{2\eta}\boldsymbol{R}^{-1}\boldsymbol{g}^\top\nabla V^*(\boldsymbol{x})\right),$$

$$E = \int_t^{t+l} e^{-\alpha(\tau-t)}\left[\boldsymbol{x}^\top \boldsymbol{Q}\boldsymbol{x} + \sum_i \int_0^{u_i} 2\eta\sigma^{-1}(\eta^{-1}s)r_i\mathrm{d}s\right]\mathrm{d}\tau, \ \mathrm{diag}(R) = (r_1,\cdots,r_m),$$

$$K = \frac{1}{2}E^2,$$

$$\dot{\boldsymbol{W}} = -\frac{\partial K}{\partial \boldsymbol{W}},$$

here $\boldsymbol{f}$ is the original dynamics described above, $\boldsymbol{g}$ is the actuator taking the form,

$$\boldsymbol{g} = \begin{pmatrix} \dfrac{x_1^n}{(s^n + x_1^n)^2} \\ 0 \end{pmatrix}.$$

The cost matrix $\boldsymbol{Q}$, $\boldsymbol{R}$ are the same as that in LQR. In the event-triggered mode, the weight $\boldsymbol{W}$ of critic NN is updated as,

$$\dot{\boldsymbol{W}} = \boldsymbol{0}, \ t \in [t_k, t_{k+1}),$$

$$\boldsymbol{W}^+ = \boldsymbol{W} - \beta\frac{\partial K}{\partial \boldsymbol{W}}, t = t_{k+1},$$

with $\beta$ being the learning rates of the weight. We initialize the weight as $\boldsymbol{W} = (W_{ij} = 4)_{2\times 2}$. We set the learning rate as $\beta = 1e-2$, the event function is set as $h = \|\boldsymbol{e}\|^2 - \frac{(1-\lambda_y^2)\underline{\lambda}(\boldsymbol{Q})}{\eta^2\lambda_x^2}\|\boldsymbol{x}\|^2$, $\lambda_y^2 = 0.6$ according to (Cheng et al., 2023). In the original work (Xue et al., 2022), historical data is considered as multiple integral on time interval $[t^j, t^j + l] \subset [t_k, t_{k+1})$ like $E = E_{[t,t+l]}$. To simplify the calculation, here we merge the multiple historical data to a single integral on time interval $[t_k, t_k + l]$ with $l = \min(l^*, t_{k+1} - t_k)$, here $l^* = 1.2$ is a predefined length of historical data. We tuned the hyperparameters in the same way with Critic-Actor ETC, and the final results are $\alpha = 0.1, \eta = 1.0, \lambda_x = 0.1, \sigma(\cdot) = \mathrm{Id}(\cdot)$.

**Test configurations.** For implementing the controller in the event-triggered mode, we set the event function for Neural ETC-PI, Quad-NLC, BALSA as

$$\nabla V \cdot (\boldsymbol{f}(\boldsymbol{x}, \boldsymbol{u}(\boldsymbol{x} + \boldsymbol{e})) - \boldsymbol{f}(\boldsymbol{x}, \boldsymbol{u}(\boldsymbol{x}))) - \sigma V(\boldsymbol{x}),$$

the event function for Neural ETC-MC as

$$\nabla V \cdot (\boldsymbol{f}(\boldsymbol{x}, \boldsymbol{u}(\boldsymbol{x} + \boldsymbol{e})) - \boldsymbol{f}(\boldsymbol{x}, \boldsymbol{u}(\boldsymbol{x}))) - \sigma\alpha(\|\boldsymbol{x}\|),$$

the event function for NLC as

$$\nabla V \cdot (\boldsymbol{f}(\boldsymbol{x}, \boldsymbol{u}(\boldsymbol{x} + \boldsymbol{e})) - \sigma\boldsymbol{f}(\boldsymbol{x}, \boldsymbol{u}(\boldsymbol{x}))),$$

the event function for LQR as (Heemels et al., 2012)

$$(\sigma - 1)(\boldsymbol{x} - \boldsymbol{P}_1)^\top \boldsymbol{Q}_1(\boldsymbol{x} - \boldsymbol{P}_1) + 2(\boldsymbol{x} - \boldsymbol{P}_1)^\top \boldsymbol{SBK}\boldsymbol{e},$$

where the $\sigma$ is set as 0.5 for all models. For the initial value, we set $\boldsymbol{x}_0 = \boldsymbol{P}_2 + \boldsymbol{\xi}_i, \boldsymbol{\xi}_i \sim \mathcal{U}[-1, 1], i = 1, \cdots, 5$, the random seed is $(2, 4, 5, 6, 7)$.

A.3.3. LORENZ SYSTEM

Here we model the state of the Lorenz system under fully actuated control $\boldsymbol{u} = (u_1, u_2, u_3)$ as $\boldsymbol{x} = (x, y, z)^\top$,

$$
\begin{aligned}
\dot{x} &= \sigma(y - x) + u_1, \\
\dot{y} &= \rho x - y - xz + u_2, \\
\dot{z} &= xy - \beta z + u_3.
\end{aligned}
$$

We aim to stabilize the zero solution of this chaotic system. We consider $\sigma = 10$, $\rho = 28$, $\beta = 8/3$. For training controller $\boldsymbol{u}$, we uniformly sample 5000 data from the state region $[-10, 10]$. We construct the controllers as follows.

**Neural ETC-PI.** We parameterize $V(\boldsymbol{x})$ as ICNN$(3, 64, 1)$, $\boldsymbol{u}(\boldsymbol{x})$ as Control$(3, 64, 64, 3)$ with $\mathcal{F} = $ ReLU. Since the Ode solver in the training process require high computational resources, we down-sample 2000 data from the original dataset for training. We set the iterations for warm up as 500, the iterations and batch size for calculating the triggering times as 100 and 10, the learning rate as lr $= 0.05$, the weight factor for event loss as $\lambda_2 = \frac{100}{2000}$.

**Neural ETC-MC.** We parameterize $V(\boldsymbol{x})$ as ICNN$(3, 64, 1)$, $\boldsymbol{u}(\boldsymbol{x})$ as Control$(3, 64, 64, 3)$. We set the iterations as $500 + 100$, the learning rate as lr $= 0.05$, the weight factor for event loss as $\lambda_2 = 0.1$.

**NLC.** We parameterize $V(\boldsymbol{x})$ as MLP$(3, 64, 64, 1)$, $\boldsymbol{u}(\boldsymbol{x})$ as MLP$(3, 64, 64, 3)$. We set the iterations as $500 + 100$, the learning rate as lr $= 0.05$, the loss function is

$$
L = \frac{1}{N} \sum_{i=1}^{N} \left[ \left( \mathcal{L}_{\boldsymbol{f}_{\boldsymbol{u}_\phi}} V_{\boldsymbol{\theta}}(\boldsymbol{x}_i) \right)^+ + (V_{\boldsymbol{\theta}}(\boldsymbol{x}_i))^+ \right] + (V_{\boldsymbol{\theta}}(\boldsymbol{0}))^+,
$$

notice that we select the last term in the right hand side as $(V_{\boldsymbol{\theta}}(\boldsymbol{0}))^+$ instead of $V_{\boldsymbol{\theta}}(\boldsymbol{0})^2$ since the former performs better than the latter. We also resample 5000 data from $[-5, 5]$ since the NLC performs poorly in the original dataset, the similar case holds for Quad-NLC.

**Quad-NLC.** We parameterize $V(\boldsymbol{x})$ as $\boldsymbol{x}^\top \text{MLP}(3, 64, 3)^\top \text{MLP}(3, 64, 3)\boldsymbol{x}$, $\boldsymbol{u}(\boldsymbol{x})$ as MLP$(3, 64, 64, 3)$. We set the iterations as $500 + 100$, the learning rate as lr $= 0.05$, the loss function is

$$
L = \frac{1}{N} \sum_{i=1}^{N} \left( \mathcal{L}_{\boldsymbol{f}_{\boldsymbol{u}_\phi}} V_{\boldsymbol{\theta}}(\boldsymbol{x}_i) + V_{\boldsymbol{\theta}}(\boldsymbol{x}_i) \right)^+ + V_{\boldsymbol{\theta}}(\boldsymbol{0})^2.
$$

**BALSA.** For this QP based method, we set the object function as

$$
\begin{aligned}
\min_{\boldsymbol{u}, d_1, d_2} &\frac{1}{2} \|\boldsymbol{u}\|^2 + p_1 d_1^2, \\
\text{s.t.} &\mathcal{L}_{\boldsymbol{f}_{\boldsymbol{u}}} V - V \leq d_1.
\end{aligned}
$$

We choose $V = \frac{1}{2}\|\boldsymbol{x}\|^2, p_1 = 20$.

**LQR.** We linearize the controlled dynamic near the zero solution as

$$
\begin{aligned}
\dot{\boldsymbol{x}} &= \boldsymbol{A}\boldsymbol{x} + \boldsymbol{B}\boldsymbol{u}, \\
\boldsymbol{A} &= \begin{pmatrix} -\sigma & \sigma & 0 \\ \rho & -1 & 0 \\ 0 & 0 & -\beta \end{pmatrix} \\
\boldsymbol{B} &= \begin{pmatrix} 1 & 0 & 0 \\ 0 & 1 & 0 \\ 0 & 0 & 1 \end{pmatrix}
\end{aligned}
$$

We set the cost matrix in LQR as

$$\boldsymbol{Q} = \begin{pmatrix} 5 & 0 & 0 \\ 0 & 10 & 0 \\ 0 & 0 & 5 \end{pmatrix},$$

$$\boldsymbol{R} = \begin{pmatrix} 0.1 & 0 & 0 \\ 0 & 0.1 & 0 \\ 0 & 0 & 0.1 \end{pmatrix}.$$

The obtained Riccati solution $\boldsymbol{S}$ forms the Lyapunov function $V = \frac{1}{2}\boldsymbol{x}^\top \boldsymbol{S}\boldsymbol{x}$, the controller is $\boldsymbol{u} = -\boldsymbol{K}\boldsymbol{x}$ where $\boldsymbol{K} \in \mathbb{R}^{3\times 3}$ is returned by the lqr solver. The Lie derivative of the Lyapunov function is $-\boldsymbol{x}^\top \boldsymbol{Q}_1 \boldsymbol{x}$ with $\boldsymbol{Q}_1 = \boldsymbol{Q} + \boldsymbol{K}^\top \boldsymbol{R}\boldsymbol{K}$.

**Critic-Actor ETC.** The updating procedure is the same as that in Appendix A.3.2, we set the hyperparameters as $\alpha_c = \alpha_a = 5e-4$, ethres $= 0.3$, we note that for the chaotic system, the event-triggered dynamics is easy to explode when $\alpha_{c,a}$ are slightly larger than $1e-3$. The actuator in this example is the identity matrix as $\boldsymbol{g} = I_{3\times 3}$.

**IRL ETC.** The updating procedure is the same as that in Appendix A.3.2, we set the hyperparameters as $\beta = 1e-2$, $\alpha = 0.1$, $\sigma(\cdot) = \tanh(\cdot)$, $\eta = 10$, $\lambda_x = 1.0$, $\lambda_y^2 = 0.6$. The actuator in this example is the identity matrix as $\boldsymbol{g} = I_{3\times 3}$.

**Test configurations.** We select the same event functions as those for GRN to implement the event-triggered control, except for setting $\sigma = 0.99$ for LQR since it fails in the case $\sigma = 0.5$. For the initial value, we randomly select 5 points in the original dataset using `numpy.random.choice` method in Python, and the random seeds are set as $\{3, 5, 7, 8, 9\}$.

### A.3.4. MICHAELIS–MENTEN MODEL

Consider the coupled subcellular model under topology control as

$$\dot{x}_i = -Bx_i + \sum_{i=1}^{100} A_{ij}\frac{x_j^2}{1+x_j^2} + \delta A_{ii}\frac{x_i^2}{1+x_i^2}.$$

This dynamic has two attractor, inactive state $\boldsymbol{P}_1 = \boldsymbol{0}$ represents the cell apoptosis and the active $\boldsymbol{P}_2$ represents the reviving cell state. We aim at regulating the cell state to the reviving state through only tuning the diagonal topology structure, which can be achieved experimentally via drugs or electrical stimulation. Therefore, an ideal control should be updated as little as possible since the frequent stimulation may do harm to the cells. For training controller $\boldsymbol{u} = (\delta A_{11}, \cdots, \delta A_{100,100})$, we uniformly sample 1000 data from the state region $[-10, 10]$. Similarly to that in GRN, we modify the parameterized $V$ and $\boldsymbol{u}$ functions s.t. $V(\boldsymbol{P}_2) = 0$, $\boldsymbol{u}(\boldsymbol{P}_2) = 0$. We construct the controllers as follows.

**Neural ETC-PI.** We parameterize $V(\boldsymbol{x})$ as ICNN$(100, 64, 1)$, $\boldsymbol{u}(\boldsymbol{x})$ as Control$(100, 64, 64, 100)$ with $\mathcal{F} = \text{ReLU}$. Since the dimension of the task is very high, the ODE solver has very high computational cost in solving the triggering times. We set the the iterations and batch size for calculating the triggering times as 10 and 5. If the readers have more powerful computing device, larger iterations and batch size are recommended. We set the iterations for warm up as 500, the learning rate as lr $= 0.01$, the weight factor for event loss as $\lambda_2 = \frac{100}{1000}$. In the case, we try a combination of Neural ETC-PI and Neural ETC-MC by setting the stabilzaition loss as

$$L_{\text{stab}} = \frac{1}{N}\sum_{i=1}^{N}\left(\mathcal{L}_{\boldsymbol{f}_{\boldsymbol{u}_{\boldsymbol{\phi}}}}V_{\boldsymbol{\theta}}(\boldsymbol{x}_i) + \alpha_{\boldsymbol{\theta}_\alpha}(\|\boldsymbol{x}_i\|)\right)^+$$

and we also penalize the Lipschitz constant of $\alpha^{-1}$ by adding term $L_{\alpha^{-1}}$ to the loss function with weight $0.1$. The dataset $\{x_i\}$ for $L_{\alpha^{-1}}$ is generated by equidistant sampling on $[0, 10]$.

**Neural ETC-MC.** We parameterize $V(\boldsymbol{x})$ as ICNN$(100, 200, 1)$, $\boldsymbol{u}(\boldsymbol{x})$ as Control$(100, 200, 200, 100)$. We set the iterations as 500, the learning rate as lr $= 0.05$, the weight factor for event loss as $\lambda_2 = 0.1$. The dataset $\{x_i\}$ for $L_{\alpha^{-1}}$ is generated by equidistant sampling on $[0, 5]$.

**NLC.** We parameterize $V(\boldsymbol{x})$ as $\text{MLP}(100, 200, 200, 1)$, $\boldsymbol{u}(\boldsymbol{x})$ as $\text{MLP}(100, 200, 200, 100)$. We set the iterations as $500$, the learning rate as $\text{lr} = 0.01$, the loss function is

$$L = \frac{1}{N} \sum_{i=1}^{N} \left[ \left( \mathcal{L}_{\boldsymbol{f}_{\boldsymbol{u}_\phi}} V_{\boldsymbol{\theta}}(\boldsymbol{x}_i) \right)^+ + (V_{\boldsymbol{\theta}}(\boldsymbol{x}_i))^+ \right] + V_{\boldsymbol{\theta}}(\mathbf{0})^2.$$

**Quad-NLC.** We parameterize $V(\boldsymbol{x})$ as $(\boldsymbol{x} - \boldsymbol{P}_2)^\top \text{MLP}(100, 200, 100)^\top \text{MLP}(100, 200, 100)(\boldsymbol{x} - \boldsymbol{P}_2)$, $\boldsymbol{u}(\boldsymbol{x})$ as $\text{MLP}(100, 200, 200, 100)$. We set the iterations as $500$, the learning rate as $\text{lr} = 0.01$, the loss function is

$$L = \frac{1}{N} \sum_{i=1}^{N} \left( \mathcal{L}_{\boldsymbol{f}_{\boldsymbol{u}_\phi}} V_{\boldsymbol{\theta}}(\boldsymbol{x}_i) + V_{\boldsymbol{\theta}}(\boldsymbol{x}_i) \right)^+ + V_{\boldsymbol{\theta}}(\mathbf{0})^2.$$

**BALSA.** For this QP based method, we set the object function as

$$\min_{\boldsymbol{u}, d_1, d_2} \frac{1}{2} \|\boldsymbol{u}\|^2 + p_1 d_1^2,$$
$$\text{s.t.} \mathcal{L}_{\boldsymbol{f}_{\boldsymbol{u}}} V - V \le d_1.$$

We choose $V = \frac{1}{2} \|\boldsymbol{x}\|^2, p_1 = 50$.

**LQR.** We linearize the controlled dynamic near the $\boldsymbol{P}_2$ solution as

$$\dot{\boldsymbol{x}} = \boldsymbol{A}(\boldsymbol{x} - \boldsymbol{P}_2) + \boldsymbol{B}\boldsymbol{u},$$
$$\rightarrow \dot{x}_i = -B + \sum_{i=1}^{100} A_{ij} \frac{2x_j^*}{(1 + (x_j^*)^2)^2} + \delta A_{ii} \frac{(x_i^*)^2}{(1 + (x_i^*)^2)^2}$$

We set the cost matrix in LQR as

$$\boldsymbol{Q} = 10\boldsymbol{I}_{100 \times 100},$$
$$R = 0.01\boldsymbol{I}_{100 \times 100}.$$

The obtained Riccati solution $\boldsymbol{S}$ forms the Lyapunov function $V = \frac{1}{2}(\boldsymbol{x} - \boldsymbol{P}_2)^\top \boldsymbol{S}(\boldsymbol{x} - \boldsymbol{P}_2)$, the controller is $\boldsymbol{u} = -\boldsymbol{K}(\boldsymbol{x} - \boldsymbol{P}_2)$ where $\boldsymbol{K} \in \mathbb{R}^{100 \times 100}$ is returned by the lqr solver. The Lie derivative of the Lyapunov function is $-(\boldsymbol{x} - \boldsymbol{P}_2)^\top \boldsymbol{Q}_1(\boldsymbol{x} - \boldsymbol{P}_2)$ with $\boldsymbol{Q}_1 = \boldsymbol{Q} + \boldsymbol{K}^\top \boldsymbol{R}\boldsymbol{K}$.

**Critic-Actor ETC.** Similarly, we set the hyperparameters as $\alpha_c = \alpha_a = 1e - 2$, $e\text{thres} = 0.2$. The actuator in this example is

$$\boldsymbol{g} = \text{diag}\left(\frac{x_1^2}{(1 + x_1^2)^2}, \cdots, \frac{x_{100}^2}{(1 + x_{100}^2)^2}\right). \tag{16}$$

Since the dynamics of $\boldsymbol{W}_c$ and $\boldsymbol{W}_a$ are both $100^2$-D, leading to a significantly high dimensional system, we reduce the dynamics as

$$V^*(\boldsymbol{x}) = \boldsymbol{W}_c^\top (x_1^2, \cdots, x_{100}^2)^\top, \ \boldsymbol{W}_c \in \mathbb{R}^{100},$$
$$\boldsymbol{u}^*(\boldsymbol{x}) = \text{diag}(\boldsymbol{W}_a)(x_1, \cdots, x_{100})^\top, \ \boldsymbol{W}_a \in \mathbb{R}^{100},$$

to the 100-D systems, for the sake of limited computational resources.

**IRL ETC.** We set the hyperparameters as $\beta = 1e - 2, \alpha = 0.1, \sigma(\cdot) = \text{Id}(\cdot), \eta = 1, \lambda_x = 0.1, \lambda_y^2 = 0.6$. The actuator in this example is

$$\boldsymbol{g} = \text{diag}\left(\frac{x_1^2}{(1 + x_1^2)^2}, \cdots, \frac{x_{100}^2}{(1 + x_{100}^2)^2}\right). \tag{17}$$

We reduce the dimension of the dynamics as the same with that in Critic-Actor ETC above.

**Test configurations.** We select the same event functions as those for GRN to implement the event-triggered control. For the initial value, we set $\boldsymbol{x}_0 = \boldsymbol{P}_1 + \boldsymbol{\xi}_i$, $\boldsymbol{\xi}_i \sim \mathcal{U}[-0.5, 0.5]$, $i = 1, c \ldots, 5$, and the random seeds are set as $\{0, 3, 4, 5, 6\}$.

### A.3.5. Motivation of selecting the benchmark systems

In (Wang et al., 2016), a geometrical approach for switching the system from ROA of one equilibrium to another, through finite changes of the experimentally feasible parameters, wherein GRN system is investigated in their paper. Since our Neural ETC has similarity to the geometrical approach in terms of adding finite non-invasive control to the system, we also study GRN in our work. The Lorenz system is a classic chaotic systems possessing plentiful shapes of dynamical trajectories, hence, the control of Lorenz (or control of chaos in a more common sense) is of important position in control literature (Ott et al., 1990; Boccaletti et al., 2000), and the control of Lorenz system under event-triggered implementation is also investigated in (Abdelrahim et al., 2015). In (Sanhedrai et al., 2022), a topological reconstruction method to the structure of complex dynamics is proposed to revive the degenerate complex system via minimal interventions, i.e., reconstructing links or nodes as small as possible, and the Michaelis–Menten model describing the evolution dynamics of sub-cellular behavior is considered as an illustration. Since the event-triggered control aims at adding feasible control to the complex system intermittently, e.g., changing the network structure slowly in time, we think it's meaningful to consider the Michaelis–Menten model in our work to see if there are essentially same parts between our method with the topological reconstruction method.

