# OpenReview forum: "Neural Event-Triggered Control with Optimal Scheduling"
_ICML.cc/2025/Conference — ICML 2025 poster_

### Official Review · Reviewer_rjuM · 2025-03-13

**Overall Recommendation:** 3

**Summary:**

This paper considers designing feedback controllers for continuous-time nonlinear systems, and the controller is updated only at certain chosen times, ensuring stability and using as few updates as possible. Experiments over three examples are provided to compare with several existing periodical control and event-trigger control techniques.

**Claims And Evidence:**

The authors claimed in the Related Work that "we are the first to study the optimization scheduling problem of ETC in the continuous dynamics". In fact, nonlinear continuous systems have been studied, for instance [1].

[1] Wang, Tengda, Guangdeng Zong, Xudong Zhao, and Ning Xu. "Data-driven-based sliding-mode dynamic event-triggered control of unknown nonlinear systems via reinforcement learning." Neurocomputing 601 (2024): 128176.

**Essential References Not Discussed:**

As mentioned previously, most recent works on neural Lyapunov control are not discussed, e.g., [3].

**Experimental Designs Or Analyses:**

First, the comparison with neural Lyapunov control is not fair. This paper uses the exponential stability condition to design the controller. The authors should compare their approach with event-triggered control work, e.g., [2], for exponential stability.

Second, the authors did not use the STOA neural Lyapunov control techniques as baseline, e.g., [3], which makes the results less convincing.

Finally, only three examples are given, and no clarification is given on how the variance is.

[2] Li, Fengzhong, and Yungang Liu. "Event-triggered stabilization for continuous-time stochastic systems." IEEE Transactions on Automatic Control 65, no. 10 (2019): 4031-4046.
[3] Yang, Lujie, Hongkai Dai, Zhouxing Shi, Cho-Jui Hsieh, Russ Tedrake, and Huan Zhang. "Lyapunov-stable neural control for state and output feedback: A novel formulation." arXiv preprint arXiv:2404.07956 (2024).

**Methods And Evaluation Criteria:**

The proposed method looks reasonable, though clarity should be improved.

**Other Comments Or Suggestions:**

None.

**Other Strengths And Weaknesses:**

The paper is very dense without clear clarification of either the high-level idea, or the each key steps.

The authors aimed to maximize the inter-event time. However, in Equation (2), the inter-event time t_{k+1}-t_k is minimized. Please clarify Equation (2).

In Sec 3.1, why is the technique called path integral? What does the optimization above Eq. (5) mean? How is Eq. (5) constructed?

Also see Theoretical Claims for other clarity issues.

**Questions For Authors:**

Please see the comments above.

**Relation To Broader Scientific Literature:**

It is restricted to the control community.

**Theoretical Claims:**

In Sec 4.4, the authors claimed that Theorem 4.2 is derived from Theorem 4.1 and Theorem 3.2. As Theorem 3.2 is in the section of the Monte Carlo approach, does it mean the result does not hold for the path integral approach?

In addition, the authors claimed stability guarantee using the Monte Carlo approach, which is very counter-intuitive, as data-driven techniques cannot provide deterministic guarantees and only statistical results can commonly be expected with some specific sampling. Please clarify how deterministic stability is assured.

---

> ### Author Rebuttal · Authors · 2025-03-31
>
> We thank the reviewer for the overall valuable comments and respond to the reviewer's major concerns one by one.
> ```
> Q1: The authors claimed in the Related Work that "we are the first to study the optimization scheduling problem of ETC in the continuous dynamics". Nonlinear continuous systems have been studied, for instance [Wang et al,2024].
> ```
> **Response**: Thank you for your feedback. Our work addresses "optimal scheduling" in event-triggered control (ETC), focusing on minimizing trigger frequency or maximizing inter-event time. Previous research typically applies reinforcement learning to optimize communication costs for discrete systems, but rarely for continuous systems[R1]. We reviewed the suggested paper [Wang et al., 2024], which considers ETC for nonlinear continuous systems. However, its objective function, shown in Equation (5), involves uncertainty bound, system state and control values, differing from our focus on the number of triggerings or inter-event time, thus not conflicting with our claims.
> ```
> Q2: The authors claimed that Theorem 4.2 is derived from Theorem 4.1 and Theorem 3.2. As Theorem 3.2 is in the section of the Monte Carlo approach, does it mean the result does not hold for the path integral approach? The authors claimed stability guarantee using the Monte Carlo approach, which is very counter-intuitive, as data-driven techniques cannot provide deterministic guarantees and only statistical results can commonly be expected with some specific sampling. Please clarify how deterministic stability is assured.
> ```
> **Response**: Thanks for your comments. Theorem 3.2 demonstrates that maximizing inter-event time corresponds to minimizing the Lipschitz constants of $\alpha^{-1}$ and $u$. We apply regularization to the Lipschitz constant of controller $u$ in both Path Integral and Monte Carlo approaches. For coherence, we positioned Theorem 3.2 in Section 3.2, although it could alternatively be placed at Section 3's start. Theorem 4.2 reveals that applying the projection operator from Theorem 4.1 to the learned controller in the post-training stage ensures optimality. This projection also works for the Path Integral approach.
>
> Regarding stability, Theorem 4.1’s projection operation $\pi$ is applied to learned controllers in the post-training stage instead of the training stage. Initially, controllers are learned using finite data in Path Integral or Monte Carlo approaches, resulting in candidate controllers $u_{\phi}$ that might not rigorously meet Lyapunov stability criteria. Then the projected controllers $\pi(u_{\phi})$ strictly fulfil the stability requirement. Therefore, this two-step process provides a rigorous stability guarantee for neural controllers.
> ```
> Q3: The comparison with neural Lyapunov control is not fair. The authors should compare their approach with event-triggered control work, e.g., [Li et al, 2019], for exponential stability.
> ```
> **Response**: Many thanks for your valuable comment. We supplement the numerical comparison with the event-triggered control method with exponential stabilization (named ETS) in the suggested paper, and we provide results in Table 1.pdf in the anonymous link https://anonymous.4open.science/r/Rebuttal_Neural-ETC-FF13/README.md (according to the Response rules of ICML2025). We note that [Li et al., 2019] assume a global Lipschitz condition (Assumption 1), which the Lorenz system does not satisfy, leading to poor performance in ETS. Additionally, their paper only designs state feedback controllers for linear systems, which is inadequate for our benchmark dynamics. Based on their theories, we developed a machine learning algorithm to derive the auxiliary function $V$ and controller $u$ that fulfil the exponential stabilization condition in Assumption 2 and Theorem 5. Our findings indicate that while ETS outperforms many benchmark methods, it remains inferior to our methods.
> ```
> Q4: The authors did not use the SOTA neural Lyapunov control techniques as the baseline, e.g., [Yang et al, 2024], which makes the results less convincing.
> ```
> **Response**: Thanks for your comment. We supplement the numerical comparison with this SOTA neural Lyapunov control technique, named as PGDNLC, in Table 1(see above link). The results show that the PGDNLC is still inferior to our methods.
> ```
> Q5: Only three examples are given, and no clarification is given on how the variance is.
> ```
> **Response**: Thanks for your careful reading and helpful comment. We have explained the motivation of selecting these three examples in Appendix A.3.5. We supplement the variance of each numerical experiment in Table 1, please refer to the above link.
>
> We would like to thank the reviewer again for his/her time and positive feedback on the paper. We hope that the reviewer will be satisfied with the responses and the supplemented results as well, and then consider revising the assessment in support of the revised paper. We may make further improvements according to your feedback.

---

> > ### Comment · Reviewer_rjuM · 2025-04-02
> >
> > Thanks for the effort.
> >
> > I have no major concerns with the experiments.

---

> > > ### Author Response · Authors · 2025-04-02
> > >
> > > We thank the reviewer for the positive feedback and support!

---

### Official Review · Reviewer_QXef · 2025-03-14

**Overall Recommendation:** 4

**Summary:**

This paper presents a novel approach to learning event-triggered controllers with maximum inter-event times using neural networks. Compared to related works, the key innovation is that the entire framework is developed for continuous dynamics and continuous triggering times. The authors propose two approaches: Neural-ETC PI and Neural-ETC MC. Both methods are based on Neural Lyapunov Control, utilizing Lyapunov functions to design triggering conditions.

The first approach, Neural-ETC PI, directly maximizes the inter-event times by integrating over the system dynamics and triggering conditions, which can be computationally expensive. The paper introduces Neural-ETC MC to address this challenge, which avoids integrating the system dynamics. This method derives a lower bound on the inter-event times under certain conditions, and the neural networks are trained to satisfy these conditions. Finally, the authors propose a projection operation to guarantee stability after training.

Through simulation experiments, the paper demonstrates that both approaches significantly outperform existing event-triggered controllers, showcasing the effectiveness of their methods.

**Claims And Evidence:**

The properties of the controller and learning method are well supported by the proofs provided in the work. While the experimental results are promising, the comparisons to the state-of-the-art are based on only five runs. Increasing the number of runs could provide more statistically significant evidence to strengthen the conclusions.

**Essential References Not Discussed:**

Ok from my point of view

**Experimental Designs Or Analyses:**

1. From the explanations of Neural-ETC PI and Neural-ETC MC, it follows that the set of controllers Neural-ETC MC can learn is a subset of those Neural-ETC PI can learn. Consequently, one might expect Neural-ETC PI to outperform Neural-ETC MC. However, according to Table 1, Neural-ETC MC significantly outperforms Neural-ETC PI and even surpasses all state-of-the-art methods by a factor of more than 100 in terms of the minimal inter-event time. This exceptional performance of Neural-ETC MC is noteworthy and could benefit from a more detailed discussion and analysis to explain the underlying reasons for these results.

2. Table 1 also includes a comparison with the method of Schlüter et al., which is an event-triggered learning approach rather than an event-triggered control method. It would be helpful to understand how this method was adapted to suit the event-triggered control setting for this comparison.

**Methods And Evaluation Criteria:**

The systems used make sense and are nicely motivated.

**Other Comments Or Suggestions:**

1. Couldn’t one eliminate l_f in the calculation of \tau_h in Theorem 3.2 and 3.3?

**Other Strengths And Weaknesses:**

1. The paper's structure is commendable, and the explanations are clear and well-presented. However, there are occasional issues with misplaced articles and minor grammatical errors—for example: "To mitigate this issue, **the** event-triggering mechanism is introduced to generate sporadic transmissions across the feedback channels of the system, compared to **the** periodic control which updates the control signal at a series of predefined explicit times.“ I recommend reviewing the manuscript for such grammatical inaccuracies, perhaps using a grammar-checking tool to enhance readability.
2. While it is understandable that space constraints might necessitate referring readers to the appendix, this is frequently done throughout the paper. Although this is acceptable for detailed proofs and hyperparameters, it also includes essential content such as more detailed discussions of related work and Algorithms 1 and 2, which are crucial for understanding the proposed methods. Including these elements in the main text would improve the paper's clarity and accessibility.
3. Furthermore, the naming of the proposed methods could be made clearer. It is somewhat unclear where the path-integral and Monte Carlo aspects come into play within the methods. Providing an explanation of these components and how they relate to the naming conventions would enhance the reader's understanding and improve the overall clarity of the paper.

**Questions For Authors:**

1. Please explain and analyze the outstanding behavior of Neural-ETC MC in detail.
2. Where exactly does Neural-ETC MC minimize the inter-event times? Is it because the approach minimizes the lipschitz constants?
3. How have you adapted Schlüter et al. to the event-triggered control setting?
4. Why exactly are your methods named Path-Integral and Monte Carlo?

**Relation To Broader Scientific Literature:**

The work demonstrates, particularly in Table 1, that the proposed Neural-ETC methods outperform the state-of-the-art on three different benchmark systems. This highlights the suitability of the approach to directly learn neural networks that minimize inter-event times.

**Theoretical Claims:**

See above

---

> ### Author Rebuttal · Authors · 2025-03-30
>
> We thank the reviewer for the overall positive feedback and the valuable comments. For the major comments, we are going to respond to them one by one.
> ```
> Q1: Please explain and analyze the outstanding behaviour of Neural-ETC MC in detail.
> ```
> **Response**: Many thanks for your valuable comment. The superiority of the Neural ETC MC over the Neural ETC PI lies in the regularization of the Lipschitz constants of function $\alpha^{-1}$ in Theorem 3.2. According to the proof of Theorem 3.2 (see Appendix A.1.2), the triggering time $t_1$ is implicitly determined by the equation $\frac{\Vert e(t_1)\Vert}{\Vert x(t_1)\Vert}=\frac{1}{P}$ initiated from $\frac{\Vert e(0)\Vert}{\Vert x(0)\Vert}=0$. Here $P$ is the tight upper bound of the Lipschitz constants of $\alpha^{-1}\circ\gamma$ in Theorem 3.2. Therefore, by minimizing the Lipschitz constants of $\alpha^{-1}$ and $\gamma$, we equivalently maximise the inter-event time. In our paper, we only consider regularization of $\alpha^{-1}$ because the Lipschitz constant of $\gamma$ is positively correlated to the Lipschitz constant of controller $u$, which is regularized in the training process.
> ```
> Q2: Where exactly does Neural-ETC MC minimize the inter-event times? Is it because the approach minimizes the Lipschitz constants?
> ```
> **Response**: Many thanks for your comments. As explained in the Response to Q1, the Neural-ETC MC minimizes the inter-event time implicitly by regularizing the Lipschitz constants of $\alpha^{-1}$ and controller $u$.
> ```
> Q3: How have you adapted Schlüter et al. to the event-triggered control setting?
> ```
> **Response**: Thanks for your careful reading. The original paper mistakenly cites the paper of Schlüter et al., we have corrected the citation as [R1][R2], here we just employ the classic LQR method with the proposed event-trigger mechanism as a baseline. The specific approach is to linearize the dynamics near the equilibrium and then apply the event-triggered LQR to stabilize the target state. We also supplemented two more methods as our baselines, one is the existing event-triggered control method for continuous dynamics and the other is the SOTA neural Lyapunov control method as suggested by Reviewer rjuM. Following the review response instruction of ICML2025, we provide the results in Table 1 at the anonymous link https://anonymous.4open.science/r/Rebuttal_Neural-ETC-FF13/README.md.
> ```
> Q4: Why exactly are your methods named Path-Integral and Monte Carlo?
> ```
> **Response**: Many thanks for your interesting comment. Our major aim is to maximize the inter-event time $t_{k+1}-t_k$ while stabilizing the dynamics. The first method calculates the inter-event time directly by integrating the temporal trajectories controlled dynamics, so we call it as a path-integral method. The second method implicitly maximizes the inter-event time by regularizing the Lipschitz constants of $\alpha^{-1}(x)$ and $u(x)$ in the loss function. The Monte Carlo estimation using $\frac{1}{N}\sum_{i}^Nu(x_i)$ and $\frac{1}{M}\sum_{j}^M\alpha^{-1}(y_j)$ over the finite training data are proportional to the Lipschitz constant of $\Vert\alpha^{-1}(x)\Vert$ and $\Vert u(x)\Vert$, respectively. Therefore, we name it as the Monte Carlo method. The first method is an explicit method, while the second one is an implicit method. So we could also rename them as Neural ETC-EXPL and Neural ETC-IMPL.
>
> ```
> Couldn’t one eliminate l_f in the calculation of \tau_h in Theorem 3.2 and 3.3?
> ```
> **Response**: Many thanks for your careful reading. Yes, we have eliminated  $l_f$ in the calculation of $\tau_h$ in Theorem 3.2 and 3.3 accordingly.
>
> **Response to Other Strengths And Weaknesses**: Many thanks for your careful reading and helpful suggestions. We have checked the manuscript and revised the typos and grammar errors. We would like to put the discussion of more related works and the algorithms into the main text if the paper could be accepted and one extra page is permitted.
>
> Finally, we would like to thank the reviewer again for his/her time and positive feedback on the paper. We may make further improvements according to your feedback.
>
> **References**
>
> [R1] Bellman, R., & Kalaba, R. E. (1965). Dynamic programming and modern control theory (Vol. 81). New York: Academic Press.
>
> [R2] Heemels, W. P., Johansson, K. H., & Tabuada, P. (2012, December). An introduction to event-triggered and self-triggered control. In 2012 ieee 51st ieee conference on decision and control (cdc) (pp. 3270-3285). IEEE.

---

> > ### Comment · Reviewer_QXef · 2025-04-04
> >
> > Thank you for your answers and the clarifications. I will increase the score to 4.

---

> > > ### Author Response · Authors · 2025-04-04
> > >
> > > We sincerely thank the reviewer for the positive feedback and support!

---

### Official Review · Reviewer_6tBp · 2025-03-18

**Overall Recommendation:** 3

**Summary:**

This study proposes a neural-based learning method for optimal scheduling in event-triggered control problems. The proposed method formulates an optimization problem to optimize the triggering rule in control problems. Then, it demonstrates how to solve this problem using neural networks. Finally, theoretical guarantees for stability and optimality are discussed.

**Claims And Evidence:**

Some parts, especially the theoretical guarantee in Theorem 4.1, may need to be discussed more carefully (please see the question section below).

**Essential References Not Discussed:**

The references in the current manuscript are sufficient.

**Experimental Designs Or Analyses:**

It may be necessary to provide more details about the gradients of $t_k$ in Section 3.1. While it explains how to compute the gradients, it is unclear to me how ODESolveEvent implements such computation.

**Methods And Evaluation Criteria:**

The proposed approach seems to be a feasible way to tackle the neural-based formulation of event-triggered control. The problem formulation in (2) appears to be natural, except for a few aspects of the objective function design (see the question section).

**Other Comments Or Suggestions:**

None.

**Other Strengths And Weaknesses:**

One of the strengths of this study is that it proposes a learning-based approach to event-triggered control problems. However, as pointed out in Experimental Designs or Analyses, it may be necessary to provide a more mathematical explanation of the computation of the derivative of $t_k$.

**Questions For Authors:**

1. I am not sure whether the values of the controller in Theorem 4.1 are continuous at the origin. This type of controller does not necessarily guarantee continuity at zero; that is, if the values of $\nabla V$ approach zero, the value of $\pi(\boldsymbol{u}, \mathcal{U}(V))$ may diverge. Such a controller achieves the goal, but it may not be acceptable in real-world applications because it requires an excessively large control input.
2. In (2), I am wondering why this objective is designed in a way that also maximizes $\|\boldsymbol{u}(\boldsymbol{x})\|_{C(\mathcal{D})}$. This leads to excessively large control values when the obtained policy is applied.

**Relation To Broader Scientific Literature:**

The contribution of this study is closely connected to control theory and control engineering. This study can be regarded as an approach to efficiently implementing event-triggered control in real complex systems.

**Theoretical Claims:**

Yes, I checked the proofs.

---

> ### Author Rebuttal · Authors · 2025-03-30
>
> We would like to thank the reviewer for the comments and valuable suggestions. We address the major concerns of the reviewer one by one.
> ```
> Q1: I am not sure whether the values of the controller in Theorem 4.1 are continuous at the origin. This type of controller does not necessarily guarantee continuity at zero; if the values of $\nabla V$ approach zero, the value of $\pi(u, U(V))$ may diverge. Such a controller achieves the goal, but it may not be acceptable in real-world applications because it requires an excessively large control input.
> ```
> **Response**: Many thanks for your comment. Actually, under the mild condition that the state space $x\in X$ is bounded and the controller $u$ is Lipschitz continuous over the state space $X$, i.e., $u\in Lip(X)$,  we can prove that the projected controller in Theorem 4.1 is continuous.  This is because that $\pi(u,U(V))\in Lip(X)\iff \frac{\max(0,\mathcal{L}\_{f_u}V)}{\Vert \nabla V\Vert^2}\cdot\nabla V\in Lip(\mathcal{X})$. Since $\Vert\frac{\nabla V}{\Vert\nabla V\Vert}\Vert$ is a continuous unit vector, and naturally is Lipschitz continuous, we only need to consider the remaining term $\frac{\max(0,\mathcal{L}\_{f_u}V-cV)}{\Vert \nabla V\Vert}$. According to the definition, all the functions occured in this term are continuous, so we only need to bound this term to obtain the Lipschitz continuity, that is, $\frac{\max(0,\mathcal{L}\_{f_u}V-cV)}{\Vert \nabla V\Vert}\in Lip(\mathcal{X})\iff\sup_{x\in\mathcal{X}}\frac{\max(0,\mathcal{L}\_{f_u}V-cV)}{\Vert \nabla V\Vert}<+\infty.$ When $\mathcal{L}\_{f_u}V\le cV$, obviously we have $\max(0,\mathcal{L}\_{f_u}V-cV)=0< +\infty$, otherwise, since $V\ge\varepsilon\Vert x\Vert^p$ and $c<0$, we have
> $
> 	\mathcal{L}\_{f-u}V-cV\ge\mathcal{L}\_{f_u}V-c\varepsilon\Vert x\Vert^p\approx\mathcal{O}(\Vert x\Vert^p)\to\infty(\Vert x\Vert\to\infty).
> $
> Thus, we have
> $
> 	\sup_{x\in\mathcal{X}}\frac{\max(0,\mathcal{L}\_{f_u}V-cV)}{\Vert \nabla V\Vert}<+\infty\iff \sup_{x\in\mathcal{X}}\Vert x\Vert<+\infty,
> $
> which completes the proof. Following the response rules of ICML2025, we provide the revised Theorem 4.1 and the proof in the anonymous link https://anonymous.4open.science/r/Rebuttal_Neural-ETC-FF13/README.md.
>
> ```
> Q2: In (2), I am wondering why this objective is designed in a way that also maximizes $\Vert u(x)\Vert_{C(\mathcal{D})}$. This leads to excessively large control values when the obtained policy is applied.
> ```
> **Response**: Thanks for your careful reading and constructive comment. It should be $\min\Vert u(x)\Vert_{C(\mathcal{D})}$ in the objective function instead of $\max\Vert u(x)\Vert_{C(\mathcal{D})}$, we have corrected this issue according to your comment. For a unified expression, we employ the objective function in Equation(2) as
> $\min_{u}\frac{1}{\min_{t_k\le T}(t_{k+1}-t_k)}+\lambda_1\Vert u(x)\Vert_{C(\mathcal{D})}$, which is consistent with the objective function in line 189.
> ```
> Q3: It may be necessary to provide a more mathematical explanation of the computation of the derivative of $t_k$.
> ```
> **Response**: Many thanks for your valuable comment. We take the computation of the derivative of $t_1$ as an example. Although in [R1] the authors provide a general mathematical expression for the derivative of $t_1$ (see Equation (9)-(11)), we can obtain a more concise expression thanks to the formulation of our event function $h(x(t),e(t))=h(x(t),x_0-x(t))$ and loss function $L=L(t_1)$ as
> $\frac{\partial t}{\partial \phi}=(\frac{\partial h}{\partial x}\frac{\partial x}{\partial t})^{-1}\frac{\partial h}{\partial x}\frac{\partial x}{\partial \phi}$. Notice that $\frac{\partial x}{\partial t}=f(x,u_{\phi}(x))$, and the Adjoint Method [R2][R3] gives the computation method for $\frac{\partial x}{\partial \theta}=(\frac{\partial x_i}{\partial \phi})\_{i=1,...,n}$ as
> $$
> a(t)=\frac{\partial x_i(t)}{\partial x(t)},
> \frac{d a(t)}{dt}=-a(t)\frac{\partial  f(x,u_{\phi})}{\partial x},
> \frac{d x_i}{d\theta}=-a(t)\frac{\partial f(x,u_{\phi})}{\partial \phi}.
> $$
> With the abovementioned mathematical expressions, we could easily calculate the derivative of $t_1$.
>
>
> We thank the reviewer again for your valuable comments. We do believe that the quality of the revised paper has been improved exceptionally from the theoretical aspect. Hopefully, the responses have sufficiently addressed the main concerns and then the reviewer will reconsider the assessment in support of the revised paper. We are looking forward to your feedback to make further improvements to the paper.
>
> **References**
>
> [R1]Chen, R. T., Amos, B., & Nickel, M. Learning Neural Event Functions for Ordinary Differential Equations. In International Conference on Learning Representations.
>
> [R2]Chen, R. T., Rubanova, Y., Bettencourt, J., & Duvenaud, D. K. (2018). Neural ordinary differential equations. Advances in neural information processing systems, 31.
>
> [R3]Pontryagin, L. S. (2018). Mathematical theory of optimal processes. Routledge.

---

> > ### Comment · Reviewer_6tBp · 2025-04-08
> >
> > Thank you very much for the clarifications.
> >
> > Regarding the proof of Theorem 4.1, the phrase
> > "Since $\left\lVert \frac{\nabla V}{\lVert \nabla V \rVert} \right\rVert$ is a continuous unit vector, and naturally is Lipschitz continuous,"
> > appears in the argument, but its validity is questionable.
> > If the region $\mathcal{D}$ contains points $x$ such that $\nabla V(x) = 0$, then $\left\lVert \frac{\nabla V(x)}{\lVert \nabla V(x) \rVert} \right\rVert$ is not well-defined if $x$ is such a point, since division by zero is undefined.
> > Therefore, $\left\lVert \frac{\nabla V}{\lVert \nabla V \rVert} \right\rVert$ is not a continuous unit vector on the region, as the region may contain points at which the expression is not well-defined.
> > The added proof may not be valid in such cases.

---

> > > ### Author Response · Authors · 2025-04-08
> > >
> > > Many thanks for your valuable comment!
> > >
> > > Regarding the singularity for $\nabla V(x)$ in the state region, according to our construction of the $V$ function in equation (5), $V$ is a strictly convex function and $\nabla V$ only takes zero value at its minimum point, i.e., the equilibrium $x^\ast$. Hence, we have:
> > >
> > > (i) The provided proof holds on region $X-\{x^\ast\}$, ensuring the continuity of $\pi(u,U(V))$ over this region.
> > >
> > > For simplicity, we consider $x^\ast=0$. Since we require the controller to vanish at the equilibrium, we have $u(0)=0$. Thus, we have $(\mathcal{L}\_{f_u}V-V)|_{x=0}=0$ and $\nabla V(0)=0$, which leads to $\pi(u,U(V))(0)=0$ at the equilibrium. Therefore, we have
> > >
> > > (ii) $\pi(u,U(V))$ is nonsingular at the equilibrium.
> > >
> > > Combining (i)(ii) together we obtain the continuity of $\pi(u, U(V))$ over the whole state region.
> > >
> > > We thank the reviewer again for the careful reading and helpful comment. Based on the above discussion, we would like to refine our proof further.
> > >
> > > We look forward to the reviewer's further feedback!

---

### Decision · Program_Chairs · 2025-05-01

**Decision:**

Accept (poster)

**Comment:**

We thank the authors for their submission.
The reviewers appreciated the novel formulation of neural event-triggered control as well as the learning-based design of the controller.
However, some parts of the paper, mentioned in the reviews, need clarification or rephrasing.
There are also some typos, and perhaps missing references that need to be addressed.